# Meta-Learning Reliable Priors in the Function Space

**Jonas Rothfuss**
ETH Zurich
`jonas.rothfuss@inf.ethz.ch`

**Dominique Heyn**
ETH Zurich
`heynd@student.ethz.ch`

**Jinfan Chen**
ETH Zurich
`georgcjf@gmail.com`

**Andreas Krause**
ETH Zurich
`krausea@ethz.ch`

## Abstract

When data are scarce, *meta-learning* can improve a learner's accuracy by harnessing previous experience from related learning tasks. However, existing methods have unreliable uncertainty estimates which are often overconfident. Addressing these shortcomings, we introduce a novel meta-learning framework, called *F-PACOH*, that treats meta-learned priors as *stochastic processes* and performs meta-level regularization directly in the *function space*. This allows us to directly steer the probabilistic predictions of the meta-learner towards high *epistemic uncertainty* in regions of insufficient meta-training data and, thus, obtain well-calibrated uncertainty estimates. Finally, we showcase how our approach can be integrated with *sequential decision making*, where reliable uncertainty quantification is imperative. In our benchmark study on meta-learning for *Bayesian Optimization (BO)*, *F-PACOH* significantly outperforms all other meta-learners and standard baselines.

## 1 Introduction

Learning new concepts and skills from a small number of examples as well as adapting them quickly in face of changing circumstances is a key aspect of human intelligence. Unfortunately, our machine learning algorithms lack such adaptive capabilities. *Meta-Learning* [1, 2] has emerged as a promising avenue towards enabling systems to learn much more efficiently by harnessing experience from previous related learning tasks [3–8]. By meta-learning probabilistic prior beliefs, we not only make more accurate predictions when given a small amount of training data, but also improve the self-assessment machine learning algorithm in the form of *epistemic uncertainty* estimates [9–12]. Such uncertainty estimates are critical for sequential decision problems such as Bayesian optimization (BO) [13, 14] and reinforcement learning [15, 16] which require efficient information gathering and exploration.

However, in most practical settings, there are only few tasks available for meta-training. Hence we face the risk of overfitting to these few tasks [17] and, consequently impairing the performance on unseen tasks. To prevent this, recent work has proposed various forms of regularization on the meta-level [18, 12, 8]. While these methods are effective in preventing meta-overfitting for the mean predictions, they fail to do so for the associated uncertainty estimates, manifested in gross *overconfidence*. Such overconfidence is highly detrimental for downstream sequential decision tasks that rely on calibrated uncertainty estimates [19, 20] to perform sufficient exploration. For instance, if the global optimum of the true target function in BO lies outside the model's 95 % confidence bounds the acquisition algorithm may never query points close to the optimum and get stuck in sub-optimal solutions. We hypothesize that previous methods yield overconfident predictions since they do not meta-regularize the predictive distribution directly. Instead, they perform meta-regularization in some latent space, for example the method parameters, which is non-trivially associated with the resulting predictive distribution and thus may not have the indented regularization effects.

35th Conference on Neural Information Processing Systems (NeurIPS 2021).

To overcome the issue of overconfident predictions in meta-learning, we develop a novel approach that regularizes meta-learned priors *directly in the function space*. We build on the PAC-Bayesian PACOH framework [12] which uses a hyper-prior over the *latent* prior parameters for meta-regularization. However, we propose to define the hyper-prior as stochastic process, characterized by its marginal distributions in the *function* space, and make the associated meta-learning problem tractable by using an approximation of the KL-divergence between stochastic processes [21]. The functional KL allows us to directly steer the meta-learned prior towards high epistemic uncertainty in regions of insufficient meta-training data and, thus, obtain reliable uncertainty estimates. When instantiating our functional meta-learning framework, referred to as *F-PACOH*, with Gaussian Processes (GPs), we obtain a simple algorithm that can be seamlessly integrated into sequential decision algorithms.

In our experiments, we showcase how *F-PACOH* can facilitate transfer and life-long learning in the context of BO, and unlike previous meta-learning methods, consistently yields well-calibrated uncertainty estimates. In our benchmark study on meta-learning for BO and hyper-parameter tuning, *F-PACOH* significantly outperforms all other meta-learners and standard baselines. Finally, we consider lifelong BO, where the meta-BO algorithm faces a sequence of BO tasks and needs build-up prior knowledge iteratively. In this challenging setting, *F-PACOH* is the only method that is able to significantly improve its optimization performance as it gathers more experience. This paves the way for exciting new applications for meta-learning and transfer such as the recurring re-optimization and calibration of complex machines and systems under changing external conditions.

## 2    Background

In this section, we formally introduce PAC-Bayesian meta-learning, the foundation of the functional meta-learning framework that we develop in Section 4. Moreover, we provide a brief description of Bayesian Optimization, which serves as the main testing ground for our proposed method.

**PAC-Bayesian Meta-Learning.** Meta-learning extracts prior knowledge (i.e., inductive bias) from a set of related learning tasks to accelerate inference in light of a new task. In the context of supervised learning, the meta-learner is given $n$ datasets $\mathcal{D}_{1,T_1}, ..., \mathcal{D}_{n,T_n}$. Each dataset $\mathcal{D}_{i,T_i} = (\mathbf{X}_i^{\mathcal{D}}, \mathbf{y}_i^{\mathcal{D}})$ consists of $T_i$ noisy function evaluations $y_{i,t} = f_i(\mathbf{x}_{i,t}) + \epsilon$ corresponding to a function $f_i : \mathcal{X} \mapsto \mathcal{Y}$ and additive noise $\epsilon$. In short, we write $\mathbf{X}_i^{\mathcal{D}} = (\mathbf{x}_{i,1}, ..., \mathbf{x}_{i,T_i})^\top$ for the matrix of function inputs and $\mathbf{y}_i^{\mathcal{D}} = (y_{i,1}, ..., y_{i,T_i})^\top$ for the vector of corresponding observations. The functions $f_i \sim \mathcal{T}$ are sampled from a task distribution $\mathcal{T}$, which can be thought of as a stochastic process that governs a random function $f : \mathcal{X} \mapsto \mathcal{Y}$. In standard Bayesian inference, we exogenously presume an – often carefully *manually designed* – prior distribution $P(h)$ over learning hypotheses $h : \mathcal{X} \mapsto \mathcal{Y}, h \in \mathcal{H}$ and combine it with empirical data to form a posterior $Q(h) = P(h|\mathcal{D})$. In contrast, in meta-learning we endogenously infer the prior $P(h)$ in a *data-driven* manner, by using the provided meta-training data.

While there exist different approaches, we focus on *PAC-Bayesian meta-learning* [22, 23, 12] due to its principled foundation in statistical learning theory. The *PACOH* framework by Rothfuss et al. [12] presumes a loss function $l(h, \mathbf{x}, y)$ and a parametric family $\{P_\phi | \phi \in \Phi\}$ of priors $P_\phi(h)$ with hyperparameter space $\Phi$. In a probabilistic setting, one typically uses the negative log-likelihood as the loss function, i.e., $l(h, \mathbf{x}, y) = -\ln p(y|h(\mathbf{x}))$. Given the meta-training data $\mathcal{D}_1, ..., \mathcal{D}_n$ and a *hyper-prior* distribution $\mathcal{P}(\phi)$ over $\Phi$, PACOH aims to infer a *hyper-posterior* distribution $\mathcal{Q}(\phi)$ over the parameters $\phi$ of the prior. Using PAC-Bayesian learning theory [24], they derive a high-probability bound on the transfer error, i.e., the generalization error for posterior inference on an unseen task $f \sim \mathcal{T}$ with priors sampled from the hyper-posterior $\mathcal{Q}(\phi)$. This meta-level generalization bound also serves as an objective for the meta-learner, minimized w.r.t. $\mathcal{Q}$:

$$J(\mathcal{Q}) = -\frac{1}{n} \sum_{i=1}^{n} \frac{1}{T_i} \mathbb{E}_{\phi \sim \mathcal{Q}} \left[ \ln Z_\beta(\mathcal{D}_{i,T_i}, P_\phi) \right] + \left( \frac{1}{\lambda} + \frac{1}{n\tilde{T}} \right) KL[\mathcal{Q}||\mathcal{P}] + \text{const.} \qquad (1)$$

Here, $\lambda > 0$, $\tilde{T} = (T_1^{-1} + ... + T_n^{-1})^{-1}$ is the geometric mean of the dataset sizes, $\ln Z(\mathcal{D}_{i,T_i}, P) := \int_{\mathcal{H}} P(h) \exp\left( -\sum_{t=1}^{T_i} l(h, \mathbf{x}_{i,t}, y_{i,t}) \right) dh$ the generalized marginal log-likelihood and $KL(\mathcal{Q}||\mathcal{P})$ the Kullback–Leibler (KL) divergence between hyper-posterior and hyper-prior. To obtain asymptotically consistent bounds, $\lambda$ is typically chosen as $\lambda = \sqrt{n}$.

**Bayesian Optimization.** Bayesian Optimization (BO) aims to find the global maximizer $\mathbf{x}^* = \arg\max_{\mathbf{x} \in \mathcal{X}} f(\mathbf{x})$ of a function $f : \mathcal{X} \to \mathbb{R}$ over a bounded domain $\mathcal{X}$. To obtain an estimate of $\mathbf{x}^*$, the BO algorithm iteratively chooses points $\mathbf{x}_1, ..., \mathbf{x}_T \in \mathcal{X}$ at which to query $f$, and observes noisy

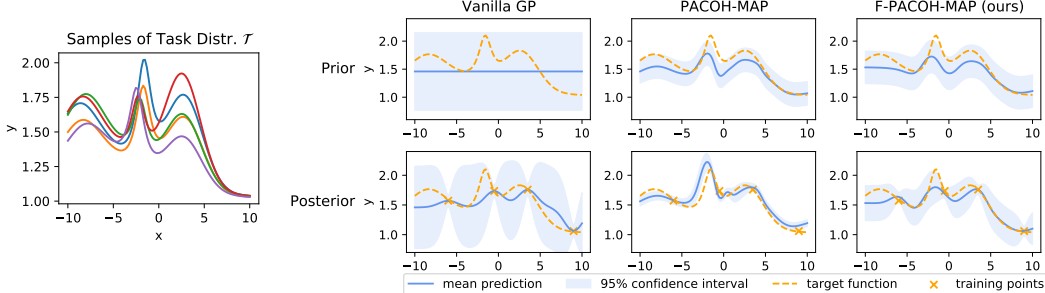

Figure 1: Prior / posterior predictions for a Vanilla GP and PACOH / F-PACOH GP meta-trained on functions from the task distribution displayed left. While PACOH yields over-confident and the Vanilla GP under-confident predictions, F-PACOH provides the well-calibrated confidence intervals.

feedback $y_1, ..., y_T \in \mathbb{R}$, e.g., via $y_t = f(\mathbf{x}_t) + \epsilon$, $\epsilon \sim \mathcal{N}(0, \sigma^2)$ with $\sigma^2 \geq$ [13, 14]. BO methods form a Bayesian *surrogate model* of the function $f$ based on previous observations $\mathcal{D}_t = \{(\mathbf{x}_t, y_t)\}$. Typically, a *Gaussian Process (GP)* $\mathcal{GP}(m(\mathbf{x}), k(\mathbf{x}, \mathbf{x}'))$ with mean $m(\mathbf{x})$ and kernel function $k(\mathbf{x}, \mathbf{x}')$ is employed to form a posterior belief $p(f(\mathbf{x})|\mathcal{D}_t)$ over function values [25]. In each iteration, the next query point $\mathbf{x}_{t+1} := \arg\max_{\mathbf{x} \in \mathcal{X}} \alpha_t(\mathbf{x})$ is chosen as maximizer of an *acquisition function* $\alpha_t(\mathbf{x})$ that is typically based on the $p(f(\mathbf{x})|\mathcal{D}_t)$ and trades-off exploration and exploitation [26–29].

## 3   Related Work

**Meta-Learning.** Common approaches in meta-learning attempt to learn a shared embedding space [3, 30–32] amortize inference by a meta-learned recurrent model [33–35] or learn the initialization of a NN so it can be quickly adapted to new tasks [6, 7, 36]. A growing body of work also uses probabilistic modeling to enable uncertainty quantification in meta-learning [9, 11, 37]. However, when only given a small number of meta-training tasks such approaches tend to overfit and fail to provide reliable uncertainty estimates. The problem of overfitting to the meta-training tasks has been brought to attention by [17, 38], followed by work that discusses potential solutions in the form of meta-regularization [8, 12, 18]. Our meta-learning framework builds on a sequence of work on PAC-Bayesian meta-learning [22, 23, 12]. However, instead of using a hyper-prior over the latent parameters of the learnable prior, we define the hyper-prior as stochastic process in the function space which gives us better control over the behavior of the predictive distributions of our meta-learned model in the absence of data.

**Learning hypotheses in the function space.** Recent work bridges the gap between stochastic processes and complex parametric models by developing a variational inference in the function space [21, 39]. Our framework in the function space heavily builds on these ideas; in particular, the proposed approximations of the KL-divergence between stochastic processes [21, 40, 41].

**Meta-Learning and Transfer for Bayesian Optimization.** To improve the sample efficiency of BO based on previous experience, a basic approach is to warm-start BO by initially evaluating configurations that have performed well on similar tasks in the past [42–44]. Another common theme is to form a global GP based model across tasks [45–48]. However, such global GPs quickly become computationally infeasible as they scale cubically in the number of tasks times the number samples per task or require hand-designed task features. Conceptually, the most similar to our work is the approach of learning a neural-network based feature map that is shared across tasks and used for BO based on a Bayesian linear regression model in the shared feature space [49, 50]. While these methods are highly scalable, they lack any form meta-level regularization and thus, in face of data-scarcity, are prone to overfitting and over-overconfident uncertainty estimates. In contrast, our method overcomes these issues thanks to its principled meta-regularization in the function space.

## 4   PAC-Bayesian Meta-Learning in the Function Space

In this section, we present our main contribution: a principled framework for meta-learning in the function space. First, we introduce the general framework, then we discuss particular instantiations and components of our approach and describe the resulting algorithm. Finally, we elaborate how our proposed meta-learner can be effectively applied in the context of Bayesian Optimization.

## 4.1 The F-PACOH Framework: Meta-Learning with Stochastic Process Hyper-Priors

We are interested in settings where the number of available meta-training tasks as well as observations may be small, and thus generalization beyond the meta-training data is challenging. In such scenarios, regularization on the meta-level plays a vital role to prevent meta-overfitting and ensure positive transfer [8, 17]. Due to its principled meta-level regularization grounded in statistical learning theory, we build on the PACOH meta-learning framework of [12] which has been introduced in Section 2.

A critical design choice when attempting to meta-learn with the PACOH objective in (1) is the hyper-prior $\mathcal{P}$ which, through the KL-divergence, becomes the dominant force that shapes the learned prior $P_\phi$ in the absence of sufficient meta-training data. Rothfuss et al. [12] define the hyper-prior as distribution over the prior parameters $\phi$ and, in particular, use a simple Gaussian $\mathcal{P}(\phi) = \mathcal{N}(\phi; 0, \sigma_{\mathcal{P}}^2 I)$. While this may act as a form of smoothness regularization on the prior, it is unclear how such a hyper-prior shapes the predictions of our meta-learned model in the function space, especially in regions where no training data is available. In such regions of data scarcity, we ideally want our meta-learned model to make conservative predictions, characterized by high epistemic uncertainty. As we can observe in Figure 1, PACOH fails to do so, yielding grossly over-confident uncertainty estimates. This is particularly problematic in sequential decision making, where data is typically non-i.i.d. and we heavily rely on well-calibrated epistemic uncertainty estimates to guide exploration.

**Hyper-prior in function space.** Aiming to make meta-learned models behave more predictably outside the support of the data, we devise a hyper-prior in the function space. We assume that the hyper-prior $\mathcal{P}$ is a stochastic process, indexed in $\mathcal{X}$ and taking values in $\mathcal{Y}$, i.e., a random function $h : \mathcal{X} \mapsto \mathcal{Y}$. Furthermore, we assume that for any finite measurement set $\mathbf{X} := [\mathbf{x}_1, ..., \mathbf{x}_k] \in \mathcal{X}^k, k \in \mathbb{N}$, the corresponding marginal distribution of function values $\rho(\mathbf{h}^{\mathbf{X}}) := \rho(h(\mathbf{x}_1), ...h(\mathbf{x}_k))$ exists and fulfills the exchangability and consistency conditions of the Kolmogorov Extension Theorem [51]. Similarly, we treat the prior $P_\phi$ as a stochastic process from which we can either sample parameters $\theta$ that correspond to functions $h_\theta : \mathcal{X} \mapsto \mathcal{Y}$ or we even have direct access to its finite marginals $p(\mathbf{h}^{\mathbf{X}}) = p(h(\mathbf{x}_1), ..., h(\mathbf{x}_k))$, e.g., multivariate normal distributions in the case of a GP. Likewise, function samples from the hyper-posterior can be obtained by hierarchical sampling ($h_\theta(\cdot)$ with $\theta \sim P_\phi, \phi \sim \mathcal{Q}$) and finite marginals by forming a mixture distribution $q(\mathbf{h}^{\mathbf{X}}) = \mathbb{E}_{\phi \sim \mathcal{Q}} \left[ p_\phi(\mathbf{h}^{\mathbf{X}}) \right]$.

Characterizing the prior, hyper-prior, and hyper-posterior as stochastic processes, we need to reconsider how to define and compute $KL[\mathcal{Q}||\mathcal{P}]$. For this purpose, we build on the result of Sun et al. [21] who show that the KL-divergence between two stochastic processes $q$ and $\rho$ can be expressed as a supremum of KL-divergences between their finite marginals:

$$KL[q, \rho] = \sup_{n \in \mathbb{N}, \mathbf{X} \in \mathcal{X}^n} KL[q(\mathbf{h}^{\mathbf{X}})||\rho(\mathbf{h}^{\mathbf{X}})] \tag{2}$$

This supremum is highly intractable, and without further specifications on the measurement set $\mathbf{X}$ it does not present a viable optimization objective [40, 41]. We thus follow a sample-based approach to computing (2), for which Sun et al. [21] provide consistency guarantees.

**Enforcing the hyper-prior.** A-priori, we want our meta-learned priors to match the structure of the hyper-prior both near the meta-training data and in regions of the domain where no data is available at all. Thus, for each task, we build measurement sets $\mathbf{X}_i = [\mathbf{X}_{i,s}^{\mathcal{D}}, \mathbf{X}_i^M]$ by selecting a random subset $\mathbf{X}_{i,s}^{\mathcal{D}}$ of the meta-training inputs $\mathbf{X}_i^{\mathcal{P}}$ as well as random points $\mathbf{X}_i^M \overset{iid}{\sim} \mathcal{U}(\mathcal{X})$ sampled independently and uniformly from the bounded domain $\mathcal{X}$. In expectation over these random measurement sets, we then compute the KL-divergence between the marginal distributions of the stochastic processes, giving us $\mathbb{E}_{\mathbf{X}_i}[KL\left[q(\mathbf{h}^{\mathbf{X}_i})||\rho(\mathbf{h}^{\mathbf{X}_i})\right]]$ as approximation of (2). Intuitively, to obtain a low expected KL-divergence, $q$ must closely resemble the behavior our stochastic process $\rho$ hyper-prior across the entire domain. Hence, in the absence of meta-training data, the hyper-prior gives us direct control over the a-priori behavior of our meta-learner in the function space. In Figure 1, we can observe how a Vanilla GP as hyper-prior shapes the predictions of our meta-learned model.

In summary, by defining the hyper-prior in the function space and using sampling-based measurement sets $\mathbf{X}_i$ to compute the functional KL divergence, we obtain the following meta-learning objective:

$$J_F(\mathcal{Q}) = \frac{1}{n} \sum_{i=1}^{n} \left( -\frac{1}{T_i} \mathbb{E}_{\phi \sim \mathcal{Q}} [\underbrace{\ln Z(\mathcal{D}_{i,T_i}, P_\phi)}_{\text{marginal log-likelihood}}] + \left( \frac{1}{\sqrt{n}} + \frac{1}{nT_i} \right) \underbrace{\mathbb{E}_{\mathbf{X}_i} \left[ KL[q(\mathbf{h}^{\mathbf{X}_i})||\rho(\mathbf{h}^{\mathbf{X}_i})] \right]}_{\text{functional KL-divergence}} \right) \tag{3}$$

In here, the marginal log-likelihood forces the meta-learned prior towards a good fit of the meta-training data while the functional KL-divergence pushes the prior towards resembling the hyper-prior's behavior in the function space. Since $J_F(\mathcal{Q})$ is inspired by the PACOH framework of [12] but works with meta-learning hypotheses in the function space, we refer to algorithms that minimize $J_F(\mathcal{Q})$ as *Functional-PACOH (F-PACOH)*.

## 4.2 Instantiations and Components of the F-PACOH framework

In the following, we discuss various instantiations of the introduced F-PACOH framework and their implications on the two main components of the meta-learning objective in (3) — the (generalized) marginal log-likelihood and the functional KL-divergence.

**Representing the hyper-posterior** $\mathcal{Q}$**.** To optimize the functional meta-learning objective $J_F$ w.r.t. $\mathcal{Q}$ we may choose a variational family of hyper-posteriors $\{\mathcal{Q}_\xi(\phi), \xi \in \Xi\}$ from which we can sample $\phi$ in a re-parameterizable manner. This allows us to obtain low-variance gradient estimates of the expectations $\nabla_\xi \mathbb{E}_{\phi \sim \mathcal{Q}_\xi}[\ln Z(\mathcal{D}_{i,T_i}, P_\phi)]$ and $\nabla_\xi \ln q_\xi(\mathbf{h^X}) = \nabla_\xi \ln \mathbb{E}_{\phi \sim \mathcal{Q}_\xi}[p_\phi(\mathbf{h^X})]$.

Alternatively, we can form a maximum a posteriori (MAP) estimate of the hyper-posterior which approximates $\mathcal{Q}$ by a Dirac delta function $\hat{\mathcal{Q}}(\phi) = \delta(\phi - \hat{\phi})$ in a single prior parameter $\hat{\phi}$. As a result, the corresponding expectations become trivial to to solve, turning (3) into a much simpler objective to minimize and making the overall meta-learning approach more practical. Thus, we focus on the MAP approximation in main body of the paper and discuss variational hyper-posteriors in Appx. A.

**The marginal log-likelihood.** In case of GPs, the marginal log-likelihood $\ln Z(\mathcal{D}_{i,T_i}, P_\phi) = \ln p(\mathbf{y}_i^\mathcal{D} | \mathbf{X}_i^\mathcal{D}, \phi)$ can be computed in closed form as (see Appx. A.1). In most other cases, e.g. when the hypothesis space $\mathcal{H} = \{h_\theta, \theta \in \Theta\}$ corresponds to the parameters $\theta$ of a neural network, we need to form an approximation of the (generalized) marginal log-likelihood $\ln p(\mathbf{y}^\mathcal{D} | \mathbf{X}^\mathcal{D}, \phi) = \ln \mathbb{E}_{\theta \sim P_\phi} \left[ e^{-\sum_{t=1}^{T_i} l(h_\theta(\mathbf{x}_{i,t}), y_{i,t})} \right]$. For further discussions on this matter, we refer to [12, 52, 53].

**Gradients of the KL-divergence.** Generally, we only require re-parametrizable sampling of functions from our prior. Following [21], we can write the gradients of the KL-divergence as

$$\nabla_\phi KL[p(\mathbf{h^X}) || \rho(\mathbf{h^X})] = \mathbb{E}_{\mathbf{h^x} \sim p_\phi} \left[ \nabla_\phi \mathbf{h^X} \left( \nabla_\mathbf{h} \ln p_\phi(\mathbf{h^X}) - \nabla_\mathbf{h} \ln \rho(\mathbf{h^X}) \right) \right] \tag{4}$$

Here, $\nabla_\phi \mathbf{h^X}$ is the Jacobian of a function sample $\mathbf{h^X}$ w.r.t. the prior parameters $\theta$. Thus, it remains to estimate the score of the prior $\nabla_\mathbf{h} \ln p_\phi(\mathbf{h^X})$ and the score of hyper-prior $\nabla_\mathbf{h} \ln \rho_\phi(\mathbf{h^X})$. In various scenarios, the marginal distributions $p(\mathbf{h^X})$ or $\rho(\mathbf{h^X})$ in the function space may be intractable and we can only sample from it. For instance, when our prior $P_\phi(\theta)$ is a known distribution over neural network (NN) parameters $\theta$, associated with NN functions $h_\theta : \mathcal{X} \mapsto \mathcal{Y}$, its marginals $p_\phi(\mathbf{h^X})$ in the function space are typically intractable. In such cases, we can use either the Spectral Stein Gradient Estimator (SSGE) [54] or sliced score matching [55] to estimate the respective score from samples. Whenever the marginal densities are available in closed form, we use automatic differentiation to compute their score. Finally, if both the prior and the hyper-prior are GPs, we use the closed-form KL-divergence between multivariate normal distributions.

## 4.3 The F-PACOH-MAP Algorithm

When focusing on the MAP approximation of the hyper-posterior, where we directly meta-learn the prior's parameter vector $\phi$, we arrive at a simple algorithm for meta-learning reliable priors in the function space. After initializing $P_\phi$, we iteratively perform stochastic gradient steps on the functional meta-learning objective $J_F(\phi)$ in (3). In each step, we iterate through all the meta-training tasks, compute the corresponding (generalized) marginal log-likelihood $\ln Z(\mathcal{D}_{i,T_i}, P_\phi)$, sample a measurement set $\mathbf{X}_i$ and estimate the gradient of the functional KL-divergence $D_{\mathrm{KL}}[p(\mathbf{h}^{\mathbf{X}_i}) || \rho(\mathbf{h}^{\mathbf{X}_i})]$ based on $\mathbf{X}_i$. In accordance with (3), we compute a weighted sum of these components and average the resulting gradients over the tasks. Finally, we use our gradient estimate $\nabla_\phi J_F(\phi)$ to perform a gradient update on $\phi$. The overall procedure, which we denote by F-PACOH-MAP, is summarized in Algorithm 1.

While the proposed function-space approach brings us many benefits, it also comes at a cost: Estimating the expectation over measurement sets $\mathbf{X}_i$ by uniform sampling and Monte Carlo estimation is subject to the curse of dimensionality. Hence, we do not expect it to work well for high-dimensional data (d > 50) such as images. For such purposes, future work may investigate alternative approximation / sampling schemes that take the data manifold into account.

**Algorithm 1** F-PACOH-MAP: Meta-Learning Reliable Priors

---

**Input:** Datasets $\mathcal{D}_{1,T_1}, ..., \mathcal{D}_{n,T_n}$, parametric family $\{P_\phi | \phi \in \Phi\}$ of priors, learning rate $\alpha$
**Input:** Stochastic process hyper-prior with marginals $\rho(\cdot)$

1: Initialize the parameters $\phi$ of prior $P_\phi$
2: **while** not converged **do**
3:     **for** $i = 1, ..., n$ **do**                 ▷ Iterate over meta-training tasks
4:         $\mathbf{X}_i = [\mathbf{X}_{i,s}^{\mathcal{D}}, \mathbf{X}_i^M]$, where $\mathbf{X}_{i,s}^{\mathcal{D}} \subseteq \mathbf{X}_i^{\mathcal{D}}, \mathbf{X}_i^M \overset{iid}{\sim} \mathcal{U}(\mathcal{X})$      ▷ Sample measurement set
5:         Estimate or compute $\nabla_\phi \ln Z(\mathbf{X}_i^{\mathcal{D}}, P_\phi)$ and $\nabla_\phi KL[p_\phi(\mathbf{h}^{\mathbf{X}_i}) || \rho(\mathbf{h}^{\mathbf{X}_i})]$
6:         $\nabla_\phi J_{F,i} = -\frac{1}{T_i} \nabla_\phi \ln Z(\mathcal{D}_{i,T_i}, P_\phi) + \left(\frac{1}{\sqrt{n}} + \frac{1}{nT_i}\right) \nabla_\phi KL[p(\mathbf{h}^{\mathbf{X}_i}) || \rho(\mathbf{h}^{\mathbf{X}_i})]$
7:     **end for**
8:     $\phi \leftarrow \phi - \alpha \frac{1}{n} \sum_{i=1}^n \nabla_\phi J_{F,i}$               ▷ Update prior parameter
9: **end while**

---

### 4.4 Application: Meta-Learning Reliable GP Priors for Bayesian Optimization

To harness the reliable uncertainty estimates of our proposed meta-learner towards improving sequential-decision making, we employ it in the context of BO. We assume that we either face a sequence of related BO problems corresponding to $n$ target functions $f_1, ..., f_n \sim \mathcal{T}$ or have access to the function evaluations from multiple such optimization runs. We use the previously collected function evaluations for meta-training with F-PACOH. Then we employ the UCB algorithm [26, 27] together with our meta-learned model to perform BO on a new target function $f \sim \mathcal{T}$.

To be able to extract sufficient prior knowledge from the data of previous BO runs, we need to choose a sufficiently rich parametrization of the GP prior $P_\phi(h) = \mathcal{GP}(h|m_\phi(x), k_\phi(x, x'))$. Hence, following [56, 38], we instantiate $m_\phi$ and $k_\phi$ as neural networks (NN), where the parameter vector $\phi$ corresponds to the weights and biases of the NN. To ensure the positive-definiteness of the kernel, we use the neural network as feature map $\Phi_\phi(x)$ on top of which we apply a squared exponential kernel.

In the standard BO setting, we would usually use a Vanilla GP with constant mean function and a conservative SE or Matérn kernel. Hence, in the absence of sufficient meta-training data, we ideally want to fall back on the behavior of such a Vanilla GP. For this reason, we use a GP prior with zero-mean and SE kernel with a small lengthscale as hyper-prior[1]. Correspondingly, the marginals $p_\phi(\mathbf{h}^\mathbf{X}) = \mathcal{N}(\mathbf{m}_{\mathbf{X},\phi}, \mathbf{K}_{\mathbf{X},\phi})$ of the prior and the hyper-prior $\rho(\mathbf{h}^\mathbf{X}) = \mathcal{N}(\mathbf{0}, \mathbf{K}_{\mathbf{X},SE})$ are multivariate normal distributions. Thus, the marginal log-likelihood (see Section A.1) and the KL-divergence are available in closed form. We provide more details in Appx. A.4 and Algorithm 3. Due to the closed-form computations, the resulting FPACOH-MAP algorithm for GPs has an asymptotic runtime complexity of $\mathcal{O}(nT^3 + nL^3)$ per iteration. In that, $T = \max_i T_i$ is the maximal number of observations in one of the meta-training datasets and $L = \max_i |\mathbf{X}_i|$ is the number of points per measurement set which can be chosen freely. In our experiment, we use $L = 20$. However, BO is most relevant when function evaluations are costly and data is scarce, i.e., both $T$ and $n$ are small. Thus, the cubic runtime is hardly a concern in practice. Alternatively, sparse GP approximations can be used to reduce the runtime complexity [57, 58]

## 5 Experiments

First, we investigate the effect of our functional meta-regularization on the uncertainty estimates of the meta-learned model. To assess the utility of the uncertainty for sequential decision making, we then evaluate our F-PACOH approach in meta-learning for BO as well as a challenging life-long BO setting. Details on the experiments can be found in Appendix B.

### 5.1 Calibration and Uncertainty Quantification

To study and illustrate our proposed meta-regularization in the function space, we use a simulated task distribution of 1-dimensional functions. Random samples from this task distribution are displayed left in Fig. 1. Using F-PACOH-MAP and PACOH-MAP [12], we meta-train a GP with $n = 10$ tasks and $T = 10$ function evaluations per task, collected with Vanilla GP-UCB [27]. PACOH-MAP corresponds to a MAP approximation of (1) with a Gaussian hyper-prior on the prior parameters $\phi$.

---

[1] Note that we standardize the inputs $\mathbf{x}$ and observations $y$ based on the empirical mean and variance of the meta-training data. The GP prior is applied in the standardized data space

Fig. 1 displays the prior and posterior predictions of the meta-learned models along those of a Vanilla GP with zero mean and SE kernel. We observe that the 95 % confidence intervals of the PACOH posterior are strongly concentrated around the mean predictions, even far away from the training points. Unlike a Vanilla GP whose confidence regions contract locally around the training points, the uncertainty estimates of PACOH contract uniformly across the domain, even far away from the training data. The fact that the true target function lies mostly outside of the confidence intervals reflects the over-confidence of PACOH-MAP. In

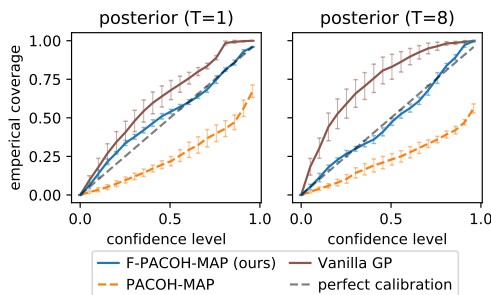

Figure 2: Calibration plots of posterior predictions corresponding to $T = 1$ and $8$ training points. Only F-PACOH yields well-calibrated predictions.

contrast, while also reflecting useful meta-learned prior knowledge, the *predictions of the F-PACOH model exhibit the behavior we desire*, i.e., small uncertainty close to the training points, higher uncertainty far away from the data.

How reliable uncertainty estimates are can be quantified by the concept of *calibration* [19, 20]. We say that a probabilistic predictor is calibrated, if, in expectation, its $\alpha\%$ confidence intervals contain $\alpha\%$ of the true function values. Fig. 2 visualizes this by plotting the fraction of the true function values covered by the posterior's confidence intervals at varying levels of $\alpha$. Here, the Vanilla-GP model is consistently under-confident, while PACOH-MAP makes grossly over-confident predictions. In contrast, F-PACOH yields well-calibrated uncertainty estimates across the entire range of confidence levels. For a more thorough analysis, Table 5 in Appendix B.5.1 reports the calibration error for all the methods and BO environments, presented Section 5.2. There, F-PACOH yields significantly lower calibration errors than the other methods in the majority of the environments. All in all, this empirically supports our claim that, through its meta-regularization in the function space, *F-PACOH is able to meta-learn priors that yield reliable uncertainty estimates*.

## 5.2 Meta-Learning for Bayesian Optimization: Setup of the Benchmark Study

We present a comprehensive benchmark study on meta-learned priors and multi-task methods for BO in which we compare our proposed method F-PACOH-MAP against various related approaches. We use the UCB aquisition algorithm [26, 27] across all the models.

**Baselines.** We compare against approaches that also meta-learn a GP prior. *PACOH-MAP* [12] is the most similar to our approach as it meta-learns a neural-network based GP prior. However, it uses a hyper-prior over prior parameters $\phi$ instead of our stochastic process formulation. Similarly, *ABLR* [50] meta-learns the feature map and prior of a Bayesian linear regression model. As a simple baseline, we meta-train a GP prior with constant mean and SE kernel by maximizing the sum of marginal log-likelihoods across tasks *(Learned GP)*. We also compare with neural processes *(NP)* [11] and rank-weighted GPs *(RWGP)* [47]. Finally, we use a *Vanilla GP* as a baseline that does not perform transfer across tasks and, if the regret definition permits[2], also report the performance of *Random Search*.

**Simulated Benchmark Environments.** We use three simulated function environments as well as three hyper-parameter optimization environments as benchmarks. Two of the simulated environments are based on the well-known *Branin* and *Hartmann6* functions for global optimization [59]. Following [60], we replace the function parameters with distributions over them in order to obtain a task distribution. Another simulated function is based on the *Camelback* function [61] which we overlay with a product of sinusoids of varying amplitude, phase and shifts along the two dimensions.

**Hyper-Parameter Optimization for machine learning algorithms.** As practical application of meta-learning for BO, we consider hyper-parameter tuning of machine learning algorithms on different datasets. In this setting, the domain $\mathcal{X}$ is an algorithm's hyper-parameter space and the target function $f(\mathbf{x})$ represents the test performance under the hyper-parameter configuration $\mathbf{x}$. The different functions $f_i \sim \mathcal{T}$ correspond to training and testing the machine learning algorithm on different datasets. The goal is to gather knowledge from hyper-parameter optimizations on different datasets so that tuning of the same algorithm on a new dataset can be done more efficiently.

---

[2]Since random search performs no model-based inference, we can only report its simple regret.

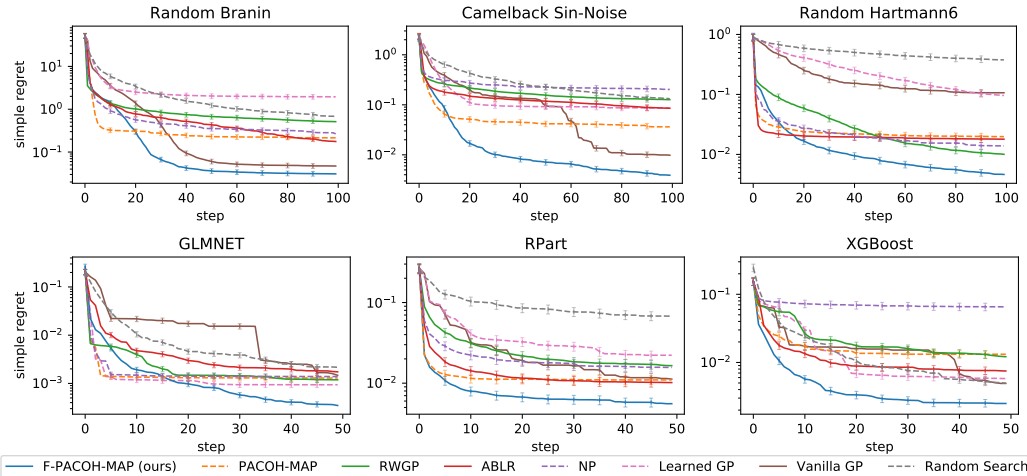

Figure 3: Performance of BO with meta-learned models on simulated function environments (top) and hyper-parameter tuning (bottom). Reported is the simple regret, averaged over seeds and function samples, alongside 95% confidence intervals. While other methods improve slowly or saturate in sub-optimal solutions, BO based on F-PACOH consistently finds near-optimal solutions quickly.

In particular, we consider three machine learning algorithms for this purpose: Generalized linear models with elastic NET regularization *(GLMNET)* [62], recursively partitioning trees *(RPart)* [63, 64] and *XGBoost* [65]. Following previous work [e.g. 50, 60], we replace the costly training and evaluation step by a cheap table lookup based on a large number of hyper-parameter evaluations [66] on 38 classification datasets from the OpenML platform [67]. In these hyper-parameter experiments, we use the area under the ROC curve (AUROC) as test metric to optimize.

## 5.3 Meta-Learning for BO: Offline Meta-Training

The first scenario we consider is where we have offline access to data of previous BO runs. In particular, using Vanilla GP-UCB, we collect $T_i$ function evaluations on $n$ tasks sampled from the task distribution $\mathcal{T}$. Depending on the dimensionality of $\mathcal{X}$ we vary $n$ and $T_i$ across the task distributions (see Tab. 1 in Appx. B). With this meta-training data $\mathcal{D}_{1,T_i}, ..., \mathcal{D}_{1,T_n}$, we meta-train F-PACOH and the considered baselines. To obtain statistically robust results, we perform independent BO runs on 10 unseen target functions / tasks and we repeat the whole meta-training & -testing process for 25 random seeds to initialize the meta-learner. To assess the performance, we report the simple regret $r_{f,t} = f(\mathbf{x}^*) - \max_{t' \le t} f(\mathbf{x}_{t'})$ as well as the inference regret $\hat{r}_{f,t} = f(\mathbf{x}^*) - f(\hat{\mathbf{x}}_t^*)$, wherein $\mathbf{x}^* = \arg\max_{\mathbf{x} \in \mathcal{X}} f(\mathbf{x})$ is the global optimum, $\mathbf{x}_t$ the point the BO algorithm chooses to evaluate in iteration $t$ and $\hat{\mathbf{x}}_t^*$ is the optimum predicted by the model at time $t$.

Fig. 3 displays the results. The inference regret is reported in Fig. 5 in Appx. B and reflects the same patterns. While PACOH-MAP yields relatively low regret after few iterations, it saturates in performance very early on and gets stuck in sub-optimal solutions. This supports our hypothesis that meta-learning with a hyper-prior in the parameter space leads to unreliable uncertainty estimates that result in poor BO solutions. Similar early saturation behavior can be observed for the other the other meta-learners, i.e., ABLR, NPs and Learned GPs. F-PACOH also yields good solutions quickly, but then *continues to improve* throughout the course of the optimization. Across all the environments, it yields *significantly lower regret* and *better final solutions* than the other methods. This demonstrates that, due to our novel meta-level regularization in the function space, F-PACOH achieves strongly positive transfer in BO without loosing the reliability and long-run performance of well-established algorithms like GP-UCB.

## 5.4 Lifelong Bayesian Optimization

Finally, we consider the scenario of lifelong BO where we face a sequence $f_0, ..., f_{N-1} \sim \mathcal{T}$ of related target functions which we want to optimize efficiently, one after another. This is a common problem setup when complex systems / machines such as the Swiss Free-Electron laser (SwissFEL) [68] that are subject to external and internal effects (e.g., drift, hysteresis). As a result, the system / machine responds differently to parameter configurations over time and needs to be

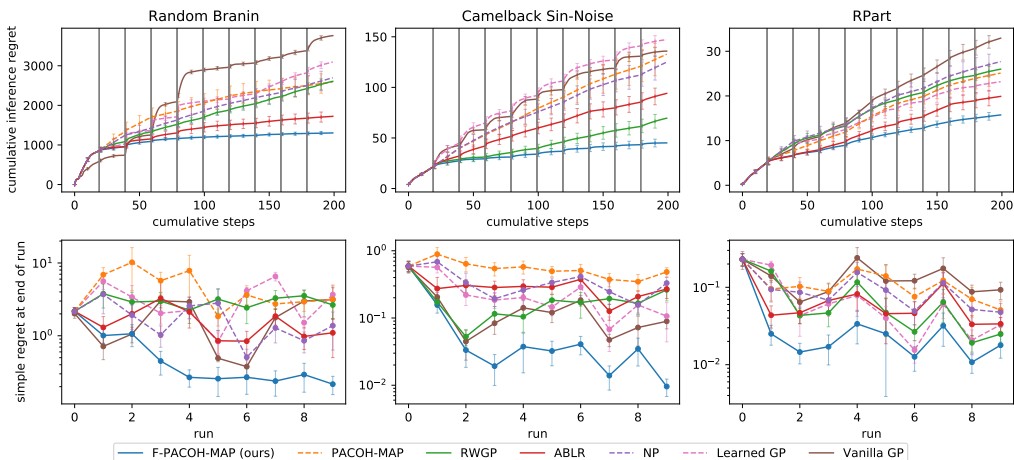

Figure 4: Lifelong BO performance on two simulated function environments and one hyper-parameter tuning benchmark (RPart). The reported results are averages over seeds and random sequences of tasks alongside 95% confidence intervals. While other meta-learners struggle to achieve positive transfer, F-PACOH is able to significantly improve the BO perfomance as it gathers more experience.

re-calibrated/optimized on a frequent basis. Our aim is to incrementally improve our performance from run to run by meta-learning with data collected in previous BO runs. This setup is more challenging than gathering meta-training offline for two reasons: 1) in the beginning, the meta-learner has much fewer meta-training tasks and 2) the meta-learned models and the collected meta-train data become interdependent, resulting in a feedback-loop between predictions and data collection on the task level. In short, this can be seen as an *exploration-exploitation trade-off on the meta-level*, which our calibrated uncertainty helps to effectively navigate.

Overall, we sequentially conduct $n = 10$ BO runs with $T_i = 20$ steps each. In the initial BO run ($i = 0$), we start without meta-training data, i.e. $\mathcal{M}_0 = \emptyset$ and thus use Vanilla GP-UCB. After each run, we add the collected function evaluations to the meta-training data, i.e., $\mathcal{M}_{i+1} = \mathcal{M}_t \cup \{D_{i,T}\}$. For the following runs ($i > 0$), we first perform meta-training with $\mathcal{M}_i$ and then run BO with the meta-learned model. As performance metric, we compute the cumulative inference regret, i.e., the sum of inference regrets of all the previous steps and runs as well as the simple regret $r_{f_i,20}$ at the end of each run. We repeat the experiment for 5 random sequences of target functions and 5 random model seeds each.

Figure 4 displays the results of our lifelong BO study for the Random Branin, Camelback Sin-Noise and RPart environment. Similar results for more environments can be found in Appx. B. At first glance, meta-learning seems to result in lower cumulative inference regret overall. However, this is mainly due to the fact that the meta-learned models start with better initial predictions which, in case of most meta-learning baselines, hardly improve within a run. Similarly, we observe that the majority of meta-learning baselines fail to consistently find better solution than a Vanilla GP. F-PACOH in stark contrast is able to *significantly improve the BO perfomance* as it gathers more experience, and *finds better solutions* by the end of a run than the other methods. This further highlights the reliability of our proposed method. Finally, the fact that F-PACOH shows strong positive transfer despite this challenging setting is highly promising, as many real-world applications may benefit from it.

## 6 Conclusion

We have introduced a novel meta-learning framework that treats meta-learned priors as stochastic processes and performs meta-level regularization directly in the function space. This gives us much better control over the behavior of predictive distribution of the meta-learned model beyond the training data. Our experiments empirically confirm that the resulting *F-PACOH* meta-learning method alleviates the major issue of over-confidence plaguing prior work, and yields well-calibrated confidence intervals. Our extensive experiments on lifelong learning for Bayesian Optimization demonstrate that *F-PACOH* is able to facilitate strong positive transfer in challenging sequential decision problems – well beyond what existing meta-learning methods are able to do. This opens many new avenues for exciting applications such as continually improving the efficiency with which we re-calibrate/-optimize complex machines and systems.

## Broader Impact

Our work focuses on meta-learning reliable priors with a small number of meta-tasks and thus a the potential to implact applications that can be cast in such as setting. Since it ensures more reliable uncertainty estimates, the proposed method is of particular interest for interactive machine learning systems that actively gather information, e.g., active learning, Bayesian optimization and reinforcement learning. For instance, the meta-learning for BO method featured in Section 5.3 and 5.4 could be applied in robotics for tuning controllers more efficiently, or in biochemistry / molecular medicine to develop novel therapeutics through protein optimization. Other potential applications include recommender systems and targeted advertisement. While improving medical applications and the control of robots promises positive impact, misuse can never be avoided.

## Acknowledgments and Disclosure of Funding

This project received funding from the Swiss National Science Foundation under NCCR Automation under grant agreement 51NF40 180545, the European Research Council (ERC) under the European Union's Horizon 2020 research and innovation program grant agreement no. 815943, and was supported by Oracle Cloud Services. Moreover, we thank Sebastian Curi and Lars Lorch for their valuable feedback.

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
