) = -\frac{1}{2}\left(\mathbf{y}^{\mathcal{D}} - \mathbf{m}_{\mathbf{X}^{\mathcal{D}},\phi}\right)^\top \tilde{\mathbf{K}}_{\mathbf{X}^{\mathcal{D}},\phi}^{-1}\left(\mathbf{y}^{\mathcal{D}} - \mathbf{m}_{\mathbf{X}^{\mathcal{D}},\phi}\right) - \frac{1}{2}\ln|\tilde{K}_{\mathbf{X}^{\mathcal{D}},\phi}| - \frac{T}{2}\ln 2\pi \quad (5)$$

where $\tilde{\mathbf{K}}_{\mathbf{X}^{\mathcal{D}},\phi} = \mathbf{K}_{\mathbf{X}^{\mathcal{D}},\phi} + \sigma^2 I$, with kernel matrix $\mathbf{K}_{\mathbf{X}^{\mathcal{D}},\phi} = [k_\phi(\mathbf{x}_l, \mathbf{x}_k)]_{l,k=1}^{T_i}$, likelihood variance $\sigma^2$, and mean vector $\mathbf{m}_{\mathbf{X}^{\mathcal{D}},\phi} = [m_\phi(\mathbf{x}_1), ..., m_\phi(\mathbf{x}_{T_i})]^\top$. In most other cases, e.g. when the hypothesis space $\mathcal{H} = \{h_\theta, \theta \in \Theta\}$ corresponds to the parameters $\theta$ of a neural network, we need to form an approximation of the (generalized) marginal log-likelihood $\ln p(\mathbf{y}^{\mathcal{D}}|\mathbf{X}^{\mathcal{D}}, \phi) = \ln \mathbb{E}_{\theta \sim P_\phi}\left[e^{-\sum_{t=1}^{T_i} l(h_\theta(\mathbf{x}_{i,t}), y_{i,t})}\right]$. For a more detailed discussion on this matter, we refer to [12, 52, 53].

## A.2   Gradients of the KL-divergence

For notational brevity, we assume a MAP approximation of the hyper-posterior so that the finite marginals of the hyper-posterior coincide with those of the prior $P_\phi$, i.e., $q(\mathbf{h}^{\mathbf{X}_i}) = p_\phi(\mathbf{h}^{\mathbf{X}_i})$. However, the concepts discussed in the remainder straightforwardly apply to the case when $\mathcal{Q}$ is a full and non-Dirac posterior distribution. In the most general case, we only require that we can sample functions from our prior in a re-parametrizable manner, i.e., there exists a map $\varphi$ and a noise distribution $p(\epsilon)$ such that for $\epsilon \sim p(\epsilon)$ we have $\mathbf{h}^{\mathbf{X}} = \varphi(\mathbf{X}, \phi, \epsilon) \sim p(\mathbf{h}^{\mathbf{X}})$. Following [21], we can derive the gradients of the KL-divergence as

$$\nabla_\phi KL[p(\mathbf{h}^{\mathbf{X}})||\rho(\mathbf{h}^{\mathbf{X}})] = \underbrace{\mathbb{E}_{\mathbf{h}^{\mathbf{X}} \sim p_\phi}\left[\nabla_\phi \ln p_\phi(\mathbf{h}^{\mathbf{X}})\right]}_{=\,0} + \mathbb{E}_\epsilon\left[\nabla_\phi \mathbf{h}^{\mathbf{X}}\left(\nabla_{\mathbf{h}} \ln p_\phi(\mathbf{h}^{\mathbf{X}}) - \nabla_{\mathbf{h}} \ln \rho(\mathbf{h}^{\mathbf{X}})\right)\right]$$

Here, the expected score of $p_\phi$ is zero and $\nabla_\phi \mathbf{h}^{\mathbf{X}} = \nabla_\phi \varphi(\mathbf{X}, \phi, \epsilon)$ is the Jacobian of $\varphi$. Thus, it remains to estimate the score of the prior $\nabla_{\mathbf{h}} \ln p_\phi(\mathbf{h}^{\mathbf{X}})$ and the score of hyper-prior $\nabla_{\mathbf{h}} \ln \rho_\phi(\mathbf{h}^{\mathbf{X}})$. In various scenarios, the marginal distributions $p(\mathbf{h}^{\mathbf{X}})$ or $\rho(\mathbf{h}^{\mathbf{X}})$ in the function space may be intractable and we can only sample from it. For instance, when our prior $P_\phi(\theta)$ is a known distribution over neural network (NN) parameters $\theta$, associated with NN functions $h_\theta : \mathcal{X} \mapsto \mathcal{Y}$, its marginals $p_\phi(\mathbf{h}^{\mathbf{X}})$ in the function space are typically intractable since the NN map is not injective w.r.t. to $\theta$. In such cases, we can use either the Spectral Stein Gradient Estimator (SSGE) [54] or sliced score matching [55] to estimate the respective score from samples. Whenever the marginal densities are available in closed form, we use automatic differentiation to compute their score. Finally, if both the prior and the hyper-prior are GPs, we use the closed-form KL-divergence between multivariate normal distributions.

## A.3   F-PACOH-VI

While we mainly focus on a maximum a-posteriori approximation of the hyper-posterior $\mathcal{Q}$ in the main paper (see Section 4.3), we now discuss the more general case of a variational approximation of the hyper-posterior.

IN this case, we presume a parametric variational family $\{\hat{\mathcal{Q}}_\xi(\phi), \xi \in \Xi\}$ of hyper-posterior distributions that are supported on the parameter space $\Phi$ of the prior. Moreover, we require that we can sample $\phi$ in a re-parameterizable manner, i.e., there exists a differentiable function $g$ and a noise distribution $p(\epsilon)$ such that $g(\xi, \epsilon) \sim \hat{\mathcal{Q}}_\xi$ for $\epsilon \sim p(\epsilon)$. An example of such variational family are Gaussians $\mathcal{N}(\phi; \mu_\mathcal{Q}, \mathrm{diag}(\sigma_\mathcal{Q}^2))$ with diagonal covariance matrices such that the variational parameters $\xi = (\mu_\mathcal{Q}, \sigma_\mathcal{Q}^2)$ coincide with the mean and the variance of the distribution. We can obtain re-parametrized samples with $\phi = g(\xi, \epsilon) = \mu_\mathcal{Q} + \sigma_\mathcal{Q}\epsilon$ and $p(\epsilon) = \mathcal{N}(0, I)$. Overall, re-parametrizable sampling allows us to obtain low-variance pathwise stochastic gradient estimators [69] of the expectation $\nabla_\xi \mathbb{E}_{\phi \sim \mathcal{Q}_\xi}[\ln Z(\mathcal{D}_{i,T_i}, P_\phi)]$ in (3).

**Algorithm 2** F-PACOH-VI: Meta-Training with GP priors
___
**Input:** Datasets $\mathcal{D}_{1,T_1}, ..., \mathcal{D}_{n,T_n}$, parametric family $\{P_\phi | \phi \in \Phi\}$ of GP priors, learning rate $\alpha$
**Input:** Parametric family of hyper-posteriors $\{\hat{\mathcal{Q}}_\xi(\phi), \xi \in \Xi\}$
**Input:** Stochastic process hyper-prior with marginals $\rho(\cdot)$
1: Initialize the parameters $\xi$ of the hyper-posterior $\mathcal{Q}_\xi(\phi)$
2: **while** not converged **do**
3:     **for** $i = 1, ..., n$ **do**                           ▷ Iterate over meta-training tasks
4:         $\mathbf{X}_i = [\mathbf{X}_{i,s}^{\mathcal{D}}, \mathbf{X}_i^M]$, where $\mathbf{X}_{i,s}^{\mathcal{D}} \subseteq \mathbf{X}_i^{\mathcal{P}}, \mathbf{X}_i^M \overset{iid}{\sim} \mathcal{U}(\mathcal{X})$    ▷ Sample measurement set
5:         $\phi \leftarrow g(\xi, \epsilon_\mathcal{Q})$ with $\epsilon_\mathcal{Q} \sim p(\epsilon_\mathcal{Q})$             ▷ Sample prior parameters from $\hat{\mathcal{Q}}_\xi$
6:         Compute $\nabla_\phi \ln Z(\mathbf{X}_i^{\mathcal{P}}, P_\phi)$ in closed form                     ▷ cf. (5)
7:         $\nabla_\mathbf{h} \ln q_\phi(\mathbf{h}^\mathbf{X}) \leftarrow \mathrm{SSGE}(q_\phi, \mathbf{h}^\mathbf{X})$           ▷ estimate hyper-posterior score
8:         $\mathbf{h}^{\mathbf{X}_i} \leftarrow \mathbf{m}_{\mathbf{X}_i,\phi} + \mathbf{K}_{\mathbf{X}_i,\phi}^{\frac{1}{2}} \epsilon_P$ with $\epsilon_P \sim \mathcal{N}(0, I)$ ▷ Sample function values from GP prior
9:         $\nabla_\xi KL[q_\xi(\mathbf{h}^{\mathbf{X}_i}) || \rho(\mathbf{h}^{\mathbf{X}_i})] \leftarrow \nabla_\xi \mathbf{h}^{\mathbf{X}_i} (\nabla_\mathbf{h} \ln q_\phi(\mathbf{h}^{\mathbf{X}_i}) - \nabla_\mathbf{h} \ln \rho(\mathbf{h}^{\mathbf{X}_i}))$
10:         $\nabla_\xi J_{F,i} = \frac{1}{T_i} \nabla_\xi g(\xi, \epsilon_\mathcal{Q}) \nabla_\phi \ln Z(\mathcal{D}_{i,T_i}, P_\phi) + \left( \frac{1}{\sqrt{n}} + \frac{1}{nT_i} \right) \nabla_\xi KL[q(\mathbf{h}^{\mathbf{X}_i}) || \rho(\mathbf{h}^{\mathbf{X}_i})]$
11:     **end for**
12:     $\xi \leftarrow \xi - \alpha \frac{1}{n} \sum_{i=1}^n \nabla_\xi J_{F,i}$                         ▷ Update hyper-posterior parameter
13: **end while**
___

A bigger challenge becomes estimating the gradient of the functional KL divergence w.r.t. the hyper-posterior parameters $\xi$, i.e.,

$$\nabla_\xi \mathbb{E}_\mathbf{X} \left[ KL[q_\xi(\mathbf{h}^\mathbf{X}) || \rho(\mathbf{h}^\mathbf{X})] \right] = \mathbb{E}_\mathbf{X} \left[ \nabla_\xi \mathbb{E}_{\mathbf{h}^\mathbf{X} \sim q_\xi} \left[ \ln q_\xi(\mathbf{h}^\mathbf{X}) - \ln \rho(\mathbf{h}^\mathbf{X}) \right] \right] \tag{6}$$

$$= \mathbb{E}_\mathbf{X} \left[ \nabla_\xi \underbrace{\mathbb{E}_{\mathbf{h}^\mathbf{X} \sim q_\xi} \left[ \ln q_\xi(\mathbf{h}^\mathbf{X}) \right]}_{-\ \mathrm{entropy}} - \nabla_\xi \underbrace{\mathbb{E}_{\mathbf{h}^\mathbf{X} \sim q_\xi} \left[ \ln \rho(\mathbf{h}^\mathbf{X}) \right]}_{-\ \mathrm{cross-entropy}} \right] \tag{7}$$

In particular, since we now have a full distribution over priors, the hyper-posterior marginals

$$q(\mathbf{h}^\mathbf{X}) = \mathbb{E}_{\phi \sim \hat{\mathcal{Q}}_\xi} \left[ p_\phi(\mathbf{h}^\mathbf{X}) \right] \tag{8}$$

are generally intractable mixing distributions, even if the prior is a GP and its marginals are tractable multivariate normal distributions. While we can still get an unbiased pathwise gradient estimator of the cross-entropy, a simple Monte Carlo gradient estimate of the negative entropy will be biased due to the concavity of the logarithm outside the expectation in $\nabla_\xi \ln q_\xi(\mathbf{h}^\mathbf{X}) = \nabla_\xi \ln \mathbb{E}_{\phi \sim \mathcal{Q}_\xi} \left[ p_\phi(\mathbf{h}^\mathbf{X}) \right]$ [53].

As discussed in Section 4.2, we may resort to score estimation techniques such as [54, 55] to obtain an estimate of KL-divergence. In particular, we rewrite the gradient of the KL-divergence as

$$\nabla_\xi KL[q_\xi(\mathbf{h}^\mathbf{X}) || \rho(\mathbf{h}^\mathbf{X})] = \mathbb{E}_{\epsilon_\mathcal{Q}, \epsilon_P} \left[ \nabla_\xi \mathbf{h}^\mathbf{X} (\nabla_\mathbf{h} \ln q_\phi(\mathbf{h}^\mathbf{X}) - \nabla_\mathbf{h} \ln \rho(\mathbf{h}^\mathbf{X})) \right] , \tag{9}$$

wherein $\nabla_\xi \mathbf{h}^\mathbf{X} = \nabla_\xi \varphi(g(\xi, \epsilon_\mathcal{Q}), \epsilon_P)$ is the Jacobian for the concatenation $g \circ \varphi$ of the reparametrization maps for the prior and the hyper-posterior. Finally, we propose to use SSGE [54] for obtaining a sampling based estimate of $\nabla_\mathbf{h} \ln q_\phi(\mathbf{h}^\mathbf{X})$. If we use a GP as hyper-prior, $\nabla_\mathbf{h} \ln \rho(\mathbf{h}^\mathbf{X})$ is the score of a multivariate normal distribution and thus available in closed form. Algorithm 2 summarizes the resulting meta-training procedure for GP priors.

### A.4 Implementation Details for F-PACOH-MAP with GPs

In the following, we provide details on our implementation of the the F-PACOH-MAP algorithm which has been introduced in Section 4.3.

**The NN-based GP prior** Following [38, 56], we parameterize the GP prior $P_\phi(h) = \mathcal{GP}(h | m_\phi(\mathbf{x}), k_\phi(\mathbf{x}, \mathbf{x}'))$, particularly the mean $m_\phi$ and kernel function $k_\phi$, as neural networks (NN). Here, the parameter vector $\phi$ corresponds to the weights and biases of the NN. To ensure the positive-definiteness of the kernel, we use the neural network as feature map $\Phi_\phi(\mathbf{x}) : \mathcal{X} \mapsto \mathbb{R}^d$ that

---

**Algorithm 3** F-PACOH-MAP: Meta-Learning GP Priors

---

    **Input:** Datasets $\mathcal{D}_{1,T_1}, ..., \mathcal{D}_{n,T_n}$, parametric family $\{P_\phi | \phi \in \Phi\}$ of GP priors, learning rate $\alpha$
    **Input:** GP hyper-prior with marginals $\rho(\mathbf{h^X}) = \mathcal{N}(\mathbf{0}, \mathbf{K_{X,\mathcal{P}}})$
 1:  Initialize the parameters $\phi$ of the GP prior $P_\phi(h) = \mathcal{GP}\left(h | m_\phi(\mathbf{x}), k_\phi(\mathbf{x}, \mathbf{x}')\right)$
 2:  **while** not converged **do**
 3:     Sample batch $I_{batch} \subseteq \{1, ..., n\}$ of $H$ tasks
 4:     **for** $i \in I_{batch}$ **do**               ▷ Iterate over meta-training tasks
 5:        $\mathbf{X}_i = [\mathbf{X}_{i,s}^{\mathcal{D}}, \mathbf{X}_i^M]$, where $\mathbf{X}_{i,s}^{\mathcal{D}} \subseteq \mathbf{X}_i^{\mathcal{D}}, \mathbf{X}_i^M \overset{iid}{\sim} \mathcal{U}(\mathcal{X})$     ▷ Sample measurement set
 6:        $\ln Z_{i,\phi} \leftarrow -\frac{1}{2}\left(\mathbf{y}_i^{\mathcal{D}} - \mathbf{m}_{\mathbf{X}_i^{\mathcal{P}},\phi}\right)^\top \tilde{\mathbf{K}}_{\mathbf{X}_i^{\mathcal{P}},\phi}^{-1}\left(\mathbf{y}_i^{\mathcal{D}} - \mathbf{m}_{\mathbf{X}_i^{\mathcal{P}},\phi}\right) - \frac{1}{2}\ln|\tilde{K}_{\mathbf{X}_i^{\mathcal{P}},\phi}| - \frac{T}{2}\ln 2\pi$
 7:        $KL_{i,\phi} \leftarrow \frac{1}{2}\left(\text{tr}\left(\mathbf{K}_{\mathbf{X}_i,\mathcal{P}}^{-1}\mathbf{K}_{\mathbf{X}_i,\phi}\right) + \mathbf{m}_{\mathbf{X}_i,\phi}^\top \mathbf{K}_{\mathbf{X}_i,\mathcal{P}}^{-1}\mathbf{m}_{\mathbf{X}_i,\phi} - L + \ln\frac{|\mathbf{K}_{\mathbf{X}_i,\mathcal{P}}|}{|\mathbf{K}_{\mathbf{X}_i,\phi}|}\right)$
 8:        $\nabla_\phi J_{F,i} = -\frac{1}{T_i}\nabla_\phi \ln Z(\mathcal{D}_{i,T_i}, P_\phi) + \left(\frac{\kappa}{\sqrt{n}} + \frac{\kappa}{nT_i}\right)\nabla_\phi KL_{i,\phi}$
 9:     **end for**
10:     $\phi \leftarrow \texttt{AdamOptimizer}(\phi, \alpha, \frac{1}{H}\sum_{i \in I_{batch}}\nabla_\phi J_{F,i})$        ▷ Update prior parameter
11: **end while**
    **Output:** Meta-learned GP prior $P_\phi(h)$

---

maps to a d-dimensional real-values feature space in which we apply a squared exponential kernel. Accordingly, the parametric kernel reads as

$$k_\phi(x, x') = \nu_P \exp\left(-||\Phi_\phi(\mathbf{x}) - \Phi_\phi(\mathbf{x}')||^2/(2l_P)\right) . \tag{10}$$

Both $m_\phi(\mathbf{x})$ and $\Phi_\phi(\mathbf{x})$ are fully-connected neural networks with 3 layers with each 32 neurons and $\tanh$ non-linearities. The kernel variance $\nu_P$ and lengthscale $l_P$ as well as the Gaussian likelihood variance $\sigma_P^2$ $p(y|h(\mathbf{x})) = \mathcal{N}(y; h(\mathbf{x}), \sigma_P^2)$ are also learnable parameters which are appended to the NN parameters $\phi$. Since $l_P$ and $\sigma_P^2$ need to be positive, we represent and optimize them in log-space.

**The hyper-prior**   We use a Vanilla GP $\mathcal{GP}(0, k_\mathcal{P}(x, x'))$ as hyper-prior stochastic process. In that,

$$k_\mathcal{P}(x, x') = \nu_\mathcal{P} \exp\left(-||x - x'||^2/(2l_\mathcal{P})\right) \tag{11}$$

is a SE kernel with variance $\nu_\mathcal{P}$ and lengthscale $l_\mathcal{P}$. Both are treated as hyper-parameters. Correspondingly, the finite marginals of the hyper-prior $\rho(\mathbf{h^X}) = \mathcal{N}(\mathbf{0}, \mathbf{K_{X,\mathcal{P}}})$ are multivariate normal distributions.

**Minimizing the functional meta-learning objective**   In case of the MAP approximation, we aim to minimize the functional meta-learning objective in (12) directly w.r.t. the prior parameters $\phi$. To minimize the objective we use mini-batching on the task level, i.e., in each iteration we sample a random subset $I_{batch} \subset \{1, ..., n\}$ with $H = |I_{batch}|$ task indices and only compute the average over the mini-batch of tasks. This stochastic estimate is unbiased and much faster to optimize than the entire sum over tasks. Since the weighting term $(1/\sqrt{n} + 1/(nT_i))$ in front of the KL divergence originates from conservative worst-case bounds on transfer error, it may be sub-optimal in expectation. Thus, following [23, 12] we add a scalar weight $\kappa > 0$ in front of it and treat it as hyper-parameter. As described in Algorithm 1, we need to also sample random measurement sets $\mathbf{X}_i$ in each iteration and for each of the tasks in the batch. In particular, we sample 10 random points $\mathbf{X}_{i,s}^{\mathcal{D}}$ without replacement from the inputs $\mathbf{X}_i^{\mathcal{D}}$ corresponding to task $i$ and sample another 10 points $\mathbf{X}_i^M \overset{iid}{\sim} \mathcal{U}(\mathcal{X})$ uniformly and independently from the bounded domain $\mathcal{X}$. The final measurement set $\mathbf{X}_i = [\mathbf{X}_{i,s}^{\mathcal{D}}, \mathbf{X}_i^M]$ is the concatenation of both sets and thus contains $L = 20$ points. Then we compute the sample-based objective

$$J_F^{MAP}(\phi) = -\frac{1}{H}\sum_{i \in I_{batch}}\left(\frac{1}{T_i}\ln Z(\mathcal{D}_{i,T_i}, P_\phi) + \left(\frac{\kappa}{\sqrt{n}} + \frac{\kappa}{nT_i}\right)KL[q(\mathbf{h^{X_i}})||\rho(\mathbf{h^{X_i}})]\right) \tag{12}$$

wherein the marginal log-likelihood (see A.1) and the KL-divergence $KL[p_\theta(\mathbf{h^X})|\rho(\mathbf{h^X})]$ are available in closed form. In particular, we use GPyTorch [70] to perfom these computations numerically stable and automatic differentiation to compute the gradients $\nabla_\phi J_F^{MAP}(\phi)$. To perform gradient updates to $\phi$, we use the adaptive learning rate method AdamW [71] with learning rate $\alpha$ and weight

| $\mathcal{T}$ | $dim(\mathcal{X})$ | $n$ | $T_i$ |
|---|---|---|---|
| Random Mixture 1d | 1 | 10 | 10 |
| Random Branin | 2 | 20 | 20 |
| Camelback Sin-Noise | 2 | 20 | 20 |
| Random Hartmann6 | 6 | 30 | 100 |
| GLMNET | 2 | 20 | 10 |
| RPart | 4 | 20 | 20 |
| XGBoost | 10 | 20 | 50 |

Table 1: Summary of meta-BO benchmark environments

decay $\omega$. In addition, we decay the learning rate every 1000 iterations by a factor $\eta \in (0,1)$. Both $\alpha$, $\omega$ and $eta$ as well as the overall number of iterations are treated are hyper-parameters. Algorithm 3 summarizes the F-PACOH-MAP meta-learning procedure for GPs.

# B Experiment Details and Further Results

## B.1 Benchmark Environments

In the following, we provide further details on benchmark environments that were used in the experiments in Section 5. Table 1 displays a summary of the environments, specifying the dimensionality of the domain, the number of meta-training tasks $n$ and the number of evaluation points $T_i$ per task used in the experiments of Section 5.1 and Section 5.3.

### B.1.1 Simulated Benchmarks

**Random Mixture Environment (1D)**    The environment corresponds to an affine combination of un-normalized Cauchy and Gaussian probability density functions:

$$p_1(x) = \frac{1}{\pi(1 + ||x - \mu_1||^2)}, \quad p_2(x) = \frac{1}{\sqrt{2\pi}} e^{-\frac{||x - \mu_2||^2}{8}}, \quad p_3(x) = \frac{1}{\pi \left(1 + \frac{||x - \mu_3||^2}{4}\right)} . \quad (13)$$

The target function follows as

$$f(x) = 2 \cdot w_1 \cdot p_1(x) + 1.5 \cdot w_2 \cdot p_2(x) + 1.8 \cdot w_3 \cdot p_2(x) + 1 \quad (14)$$

wherein the mixing weights $w_1, w_2, w_3$ are sampled independently from $\mathcal{U}(0.6, 1.4)$ and the location parameters are sampled from the Gaussians

$$\mu_1 \sim \mathcal{N}(-2, 0.3^2), \quad \mu \sim \mathcal{N}(3, 0.3^2), \quad \mu_3 \sim \mathcal{N}(-8, 0.3^2) . \quad (15)$$

The domain is the one dimensional interval $\mathcal{X} = [-10, 10]^\top$. Function samples from the environment are illustrated in Fig. 1.

**Random Branin**    The environment corresponds to random Branin functions [59] with the 2-dimensional cube $\mathcal{X} = [-5, 10] \times [0, 15]$ as domain. We denote $\mathbf{x} = (x_1, x_2)^\top$. Since we phrase BO as maximization problem, we used the negative Branin function:

$$f(x_1, x_2) = - \left(a(x_2 - bx_1^2 + cx_1 - r)^2 + s(1 - t)\cos(x_1) + s\right) \quad (16)$$

In that, the parameters $a, b, c, r, s, t$ are sampled from uniform distributions, in particular,

$$\begin{aligned} a \sim \mathcal{U}(0.5, 1.5), \quad b \sim \mathcal{U}(0.1, 0.15), \quad c \sim \mathcal{U}(1, 2), \\ r \sim \mathcal{U}(5, 7), \quad s \sim \mathcal{U}(8, 12), \quad t \sim \mathcal{U}(0.03, 0.05) . \end{aligned} \quad (17)$$

**Camelback Sin-Noise**    The environment corresponds to a Camelback function [61]

$$g(x_1, x_2) = \max\left(-(4 - 2.1 \cdot x_1^2 + x_1^4/3) * x_1^2 - x_1 x_2 - (4 \cdot x_2^2 - 4) * x_2^2, \; -2.5\right) . \quad (18)$$

plus random sinusoid functions, defined over the 2-dimensional cube $\mathcal{X} = [-2, 2] \times [-1, 2]$ as domain. Specifically, the target function is defined as

$$f(x_1, x_2) = g(x_1, x_2) + a\sin(\omega_1 * (x_1 - \rho_1))\sin(\omega_2 * (x_2 - \rho_2)) \quad (19)$$

wherein the parameters are sampled independently as

$$a \sim \mathcal{U}(0.3, 0.5), \quad \omega_1, \omega_2 \sim \mathcal{U}(0.5, 1.0), \quad \rho_1, \rho_2 \sim \mathcal{N}(0, 0.3^2) . \quad (20)$$

**Random Hartmann6** The environment corresponds to a negated and randomized version Hartmann-6D function [59] with the hyper-cube $\mathcal{X} = [0,1]^6$ as domain. In particular, the target function is defined as

$$f(\mathbf{x}) = \frac{1}{3.322368} \sum_{i=1}^{4} \alpha_i \exp\left(-\sum_{j=1}^{6} A_{i,j}(x_j - P_{i,j})^2\right) \text{, where} \qquad (21)$$

$$\mathbf{A} = \begin{pmatrix} 10.00 & 3.00 & 17.00 & 3.50 & 1.70 & 8.00 \\ 0.05 & 10.00 & 17.00 & 0.10 & 8.00 & 14.00 \\ 3.00 & 3.50 & 1.70 & 10.00 & 17.00 & 8.00 \\ 17.00 & 8.00 & 0.05 & 10.00 & 0.10 & 14.00 \end{pmatrix} \text{ and} \qquad (22)$$

$$\mathbf{P} = 10^{-4} \begin{pmatrix} 1312 & 1696 & 5569 & 124 & 8283 & 5886 \\ 2329 & 4135 & 8307 & 3736 & 1004 & 9991 \\ 2348 & 1451 & 3522 & 2883 & 3047 & 6650 \\ 4047 & 8828 & 8732 & 5743 & 1091 & 381 \end{pmatrix}. \qquad (23)$$

The parameters $\alpha_1, ..., \alpha_4$ are sampled independently from uniform distributions

$$\alpha_1 \sim \mathcal{U}(0.5, 1.5), \quad \alpha_2 \sim \mathcal{U}(0.6, 1.4), \quad \alpha_3 \sim \mathcal{U}(2.0, 3.0), \quad \alpha_4 \sim \mathcal{U}(2.8, 3.6). \qquad (24)$$

### B.1.2 Hyper-Parameter Tuning on OpenML Datasets

In our BO empirical benchmark studies, we consider use case of hyper-parameter tuning for machine learning algorithm. In particular, we consider three machine learning algorithms for this purpose:

- Generalized linear models with elastic NET regularization *(GLMNET)* [62]
- Recursively partitioning trees *(RPart)* [63, 64]
- Gradient boosting *(XGBoost)* [65]

Following previous work [e.g. 50, 60], we replace the costly training and evaluation step by a cheap table lookup based on a large number of hyper-parameter evaluations [66] on 38 classification datasets from the OpenML platform [67]. The hyper-parameter evaluations are available under a Creative Commons BY 4.0 license and can be downloaded here[3]. In effect, $\mathcal{X}$ is a finite set, corresponding to 10000-30000 random evaluations hyper-parameter evaluations per dataset and machine learning algorithm. Since the sampling is quite dense, for the purpose of empirically evaluating the meta-learned models towards BO, this finite domain can be treated like a continuous domain. All datasets correspond to binary classification. The target function we aim to optimize is the area under the ROC curve (AUROC) on a test split of the respective dataset.

We randomly split the available tasks (i.e. train/test evaluations on a specific dataset) into a set of meta-train and meta-test tasks. In the following, we list the corresponding OpenML dataset identifiers:

- meta-train tasks: 3, 1036, 1038, 1043, 1046, 151, 1176, 1049, 1050, 31, 1570, 37, 4134, 1063, 1067, 44, 1068, 50, 1461, 1462
- meta-test tasks: 335, 1489, 1486, 1494, 1504, 1120, 1510, 1479, 1480, 333, 1485, 1487, 334

Since some of machine learning algorithm's hyper-parameters were sampled by [66] in log-space, we transform the respective hyper-parameters accordingly and also adjust them to standard normal values ranges such that we can expect a reasonably good performance of a Vanilla GP with SE kernel. The hyper-parameters and transformations are listed in Table 2.

## B.2 Evaluation Methodology and Metrics

### B.2.1 Supervised Learning: Regression

**Methodology** In the following, we describe our experimental methodology and provide details on how the empirical results reported in Table 4 and Table 5 were generated. Overall, evaluating a

---

[3]https://doi.org/10.6084/m9.figshare.5882230.v2

| algorithm | hyper-parameter | type | transformation |
|-----------|-----------------|------|----------------|
| GLMNET | alpha | numeric | - |
| | lambda | numeric | $t(x) = \log_2(x)/10$ |
| RPart | cp | numeric | $t(x) = 4x$ |
| | maxdepth | integer | $t(x) = x/10$ |
| | minbucket | integer | $t(x) = x/20$ |
| | minsplit | integer | $t(x) = x/20$ |
| XGBoost | nrounds | integer | $t(x) = (x - 2000)/1000$ |
| | eta | numeric | $t(x) = (\log_2(x) + 5)/2$ |
| | lambda | numeric | $t(x) = \log_2(x)/5$ |
| | alpha | numeric | $t(x) = \log_2(x)/5$ |
| | subsample | numeric | $t(x) = (x - 0.5)/2$ |
| | booster | $\{-1, 1\}$ | -1 for 'linear' and 1 for 'tree' |
| | max_depth | integer | - |
| | min_child_weight | numeric | $t(x) = (x - 50)/20$ |
| | colsample_bytree | numeric | - |
| | colsample_bylevel | numeric | - |

Table 2: Hyper-parameters and corresponding parameter transformations for the three machine learning algorithms considered for our hyper-parameter tuning benchmark

meta-learner consists of two phases, *meta-training* and *meta-testing*. In meta-training, we perform meta=learning based on a set of datasets $\{\mathcal{D}_1, ..., \mathcal{D}_n\}$ corresponding to tasks sampled from the task distribution $\mathcal{T}$. In meta-learned model receives multiple of unseen test tasks consisting of each a train set $\mathcal{D}_{train}$ and a test set $\mathcal{D}_{test}$ that both correspond to the same function $f \sim \mathcal{T}$. The train set $\mathcal{D}_{train}$ is used to perform inference / training with the model. Then the following evaluation metrics are computed on $\mathcal{D}_{test}$.

**Log-Likelihood**   Following [72], we report the average predictive log-likelihood of test points:

$$\text{LL} = \frac{1}{|\mathcal{D}_{test}|} \sum_{(\mathbf{x},y) \in \mathcal{D}_{test}} \ln \hat{p}(y|\mathbf{x}, \mathcal{D}_{train}) \tag{25}$$

In that, $\hat{p}(\cdot|\mathbf{x})$ denotes is the predictive distribution of the respective (meta-learned) model, trained on $\mathcal{D}_{train}$ at meta-test time.

**Calibration Error**   The concept of calibration applies to probabilistic predictors that, given a new target input $\mathbf{x}_j$, produce a probability distribution $\hat{p}(y_j|\mathbf{x}_j)$ over predicted target values $y_j$ [19, 20].

Corresponding to the predictive density, we denote a predictor's cumulative density function (CDF) as $\hat{F}(y_j|\mathbf{x}_j) = \int_{-\infty}^{y_j} \hat{p}(y|\mathbf{x}_i)dy$. For confidence levels $0 \le q_h < ... < q_H \le 1$, we can compute the corresponding empirical frequency

$$\hat{q}_h = \frac{|\{y_j \mid \hat{F}(y_j|\mathbf{x}_j) \le q_h, j = 1, ..., m\}|}{m} , \tag{26}$$

based on the test dataset $\mathcal{D}_{test} = \{(\mathbf{x}_i, y_i)\}_{i=1}^m$ of $m$ samples. If we have calibrated predictions we would expect that $\hat{q}_h \to q_h$ as $m \to \infty$. Similar to [20], we can define the calibration error as a function of residuals $\hat{q}_h - q_h$, in particular,

$$\text{calib-err} = \sqrt{\frac{1}{H} \sum_{h=1}^{H} (\hat{q}_h - q_h)^2} . \tag{27}$$

Note that we while [20] reports the average of squared residuals $|\hat{q}_h - q_h|^2$, we report its square root in order to preserve the units and keep the calibration error easier to interpret. In our experiments, we compute (27) with $M = 20$ equally spaced confidence levels between 0 and 1.

### B.2.2 Offline Meta-Learning for BO

In this section, we describe the experimental methodology of the meta-learning for BO experiments in Section 5.3.

Unlike previous work [e.g. 50, 60] which collects meta-training data by uniformly sampling data from the domain, we collect meta-training data by running Vanilla GP-UCB on a the respective meta-training tasks. This is a more realistic setup since in real-wold applications we most likely only have access to non-i.i.d. data that originates from previous optimization attempts. The number of tasks $n$ and evaluations per task $T_i$ are specified in Table 1. In case of the simulated experiments, the tasks are sampled i.i.d. from the task distribution while in hyper-parameter optimization study they corresponds to the meta-train tasks listed in Appendix B.1.2.

With the collected datasets, we perform meta-training and then employ the meta-learned model towards BO with the UCB acquisition function

$$\alpha_t(\mathbf{x}) = \hat{\mu}_{t-1}(\mathbf{x}) + 2\hat{\sigma}_{t-1}(\mathbf{x}) \tag{28}$$

on unseen meta-test tasks, i.e. new functions $f \sim \mathcal{T}$. In that, $\hat{\mu}_{t-1}(\mathbf{x})$ and $\hat{\sigma}_{t-1}(\mathbf{x})$ denote the mean and standard deviation of predictive distribution $\hat{p}(y|\mathbf{x}, \{(\mathbf{x}_{t'}, y_{t'}\}_{t'=1}^{t-1})$ of the meta-learned model, given the previous BO evaluations $\{(\mathbf{x}_{t'}, y_{t'}\}_{t'=1}^{t-1}$ in this run as training data. To obtain statistically robust results, we evaluate the BO performance on 10 test tasks and repeat the entire meta-training and BO process for 25 model seeds.

To assess the BO performance, we report the *simple regret*

$$r_{f,t} = f(\mathbf{x}^*) - \max_{t' \le t} f(\mathbf{x}_{t'}) , \tag{29}$$

wherein $\mathbf{x}^* = \arg\max_{\mathbf{x} \in \mathcal{X}} f(\mathbf{x})$ is the global optimum of the target function, $\mathbf{x}_t$ the point the BO algorithm chooses to evaluate in iteration $t$. Moreover, we report the *inference regret*

$$\hat{r}_{f,t} = f(\mathbf{x}^*) - f(\hat{\mathbf{x}}_t^*) , \tag{30}$$

where $\hat{\mathbf{x}}_t^* = \arg\max_{\mathbf{x} \in \mathcal{X}} \hat{\mu}_{t-1}(\mathbf{x})$ the predicted maximum at time $t$.

### B.2.3 Lifelong BO

Unlike in the offline meta-learning setting, in the lifelong BO experiment of Section 5.4, $n = 10$ BO runs with $T_i = 20$ steps each are performed sequentially and meta-training happens online fashion after every BO run. Since, initially, there is no meta-training data available, i.e. $\mathcal{M}_0 = \emptyset$, we use a Vanilla GP as model in the first BO run. After each run, we add the collected function evaluations to the meta-training data, i.e., $\mathcal{M}_{i+1} = \mathcal{M}_t \cup \{D_{i,T}\}$. For the following runs ($i > 0$), we first perform meta-training with $\mathcal{M}_i$ and then run BO with the UCB acquisition function and the meta-learned model. In case of the simulated environments, the 10 tasks are sampled i.i.d from $\mathcal{T}$, whereas in the hyper-parameter tuning setting, we use randomly shuffled sequences of tasks from the meta-test tasks listed in Appendix B.1.2. We repeat the whole lifelong BO process for 5 random task sequences and and 5 model seeds each.

To assess the overall performance of the meta-learned to the end of lifelong BO, we compute the *cumulative inference regret*

$$R_{i,t} = \sum_{i' < i} \sum_{t'=0}^{T_{i'}} \hat{r}_{f_{i'},t'} + \sum_{t'=0}^{t} \hat{r}_{f_i,t'} , \tag{31}$$

that is, is the sum of inference regrets of all the previous steps and runs. In addition, we report the *simple regret $r_{f_i,20}$ at the end of each BO run.* While the former metric gives us a good picture of the respective meta-learner throughout the course of the lifelong Bayesian Optimization, the latter metric tells us what is the best point found per run and how this end-of-run solution develops as the meta-learned collects more meta-training tasks.

### B.3 Open Source Code, Experiment Data and Compute Resources

We provide source code which includes an implementation of F-PACOH, the baselines, the environments as well as the experiment scripts to reproduce the presented empirical results. The source code is part of our code and data repository which can be accessed via:

The repository also includes the meta-training data which has been collected with GP-UCB as well as detailed recordings of our experiments that were the basis for the plots and tables presented in this paper.

All experiments for this work, especially the hyper-parameter sweeps for F-PACOH and the baselines were conduced on CPU-only machines on Oracle Cloud. Overall, we have used 192,428 OCPU hours, an equivalent of 125 days on a 64-core machine.

## B.4 Hyper-Parameter Selection for F-PACOH-MAP and the baselines

|  | symbol | sampling type | value range / choices |
|---|---|---|---|
| learning rate | $\alpha$ | loguniform | [0.0001, 0.005] |
| learning rate decay | $\eta$ | loguniform | [0.8, 1.0] |
| weight decay | $\omega$ | loguniform | [0.00001, 0.1] |
| task batch size | $H$ | choice | {4, 10} |
| number of meta-training iterations | - | choice | {2000, 4000, 8000} |
| hyper-prior lengthscale | $l_{\mathcal{P}}$ | loguniform | [0.1, 1.0] |
| hyper-prior factor | $\kappa$ | loguniform | [0.0001, 0.5] |
| kernel feature dimensionality | $d$ | choice | {2, 6} |

Table 3: Hyper-parameter search ranges and uniform sampling types for F-PACOH-MAP

The choose the hyper-parameters of F-PACOH-MAP and the considered baselines we use random search. The hyper-parameters are either sampled uniformly from a finite set of choices or sampled log-uniformly over a range. The particular choice sets and ranges for F-PACOH-MAP are listed in Table 3. We draw 128 random hyper-parameter samples, employ the respective method on a specific environment and select the hyper-parameters corresponding to the best hyper-parameter settings employed on three validation tasks that are distinct from the meta-test tasks. Specifically, for the offline meta-learning experiment, we select the best hyper-parameters based on the last simple regret $r_{f,T}$ and the average inference regret during that last 50 iterations $\frac{1}{50} \sum_{t=T-50}^{T} \hat{r}_{f,T}$. In the life-long BO setting, we use the average inference regret over the last 5 steps per run as well as the last simple regret per run, averaged over all runs, as performance metrics. In particular, we rank the 128 hyper-parameter runs for each of the two metrics and choose the hyper-parameter setting with the highest average ranking. We perform this random hyper-parameter search for all the baselines and all the environments individually.

The resulting hyper-parameter configurations for all the methods and environments are reported in our experiment repository and can be accessed / downloaded under the following link: 

## B.5 Further Experiment Results

### B.5.1 Supervised Meta-Learning Experiments

Following the methodology described in Appendix B.2.1, we present a meta-learning benchmark study that evaluates the F-PACOH method as well as multiple other baselines based on their supervised learning predictions. Unlike the empirical studies in Section 5.3 and 5.4 which evaluate the BO performance of the meta-learned, this study only considers regression. We use the task data which was collected using GP-UCB as part of the experiment in Section 5.3 for meta-training and meta-testing. See Table 1 for a summary of the number of tasks and data points per environment. In particular, we use half of the $n$ tasks for meta-training and the other half for meta-testing. For the meta-test tasks, we use 50% of the points for inference and the other 50% for computing the test metrics.

Table 4 reports the average test log-likelihood and Table 5 lists the corresponding calibration error. Overall, we observe that, alongside ABLR, F-PACOH achieves the best test log-likelihood across the environments. In terms of the calibrations of its uncertainty estimates, F-PACOH significantly outperforms the other methods in the majority of the environments. This is consistent with our

| | Rand. Branin | Camelb. Sin-Noise | Rand. Hartmann6 | GLMNET | RPart | XGBoost |
|---|---|---|---|---|---|---|
| FPACOH-MAP | $\mathbf{-1.854 \pm 0.015}$ | $-0.235 \pm 0.044$ | $\mathbf{1.448 \pm 0.044}$ | $\mathbf{1.692 \pm 0.041}$ | $1.596 \pm 0.087$ | $1.051 \pm 0.079$ |
| PACOH-MAP | $-2.507 \pm 0.267$ | $-0.716 \pm 0.029$ | $1.337 \pm 0.048$ | $1.369 \pm 0.025$ | $-0.808 \pm 0.217$ | $0.916 \pm 0.027$ |
| ABLR | $-3.684 \pm 0.006$ | $-0.738 \pm 0.008$ | $1.358 \pm 0.059$ | $1.233 \pm 0.012$ | $-0.471 \pm 0.209$ | $0.949 \pm 0.061$ |
| NP | $-4.621 \pm 0.232$ | $-0.888 \pm 0.031$ | $1.288 \pm 0.099$ | $0.875 \pm 0.523$ | $-0.566 \pm 0.496$ | $0.421 \pm 0.144$ |
| Learned GP | $-3.738 \pm 0.003$ | $\mathbf{0.395 \pm 0.004}$ | $1.305 \pm 0.001$ | $1.412 \pm 0.002$ | $\mathbf{1.611 \pm 0.002}$ | $\mathbf{1.587 \pm 0.004}$ |
| Vanilla GP | $-3.027 \pm 0.000$ | $0.170 \pm 0.000$ | $1.351 \pm 0.041$ | $0.831 \pm 0.000$ | $1.380 \pm 0.000$ | $0.872 \pm 0.000$ |

Table 4: Average test log-likelihood of various meta-learned models as well as a Vanilla GP on meta-test tasks generated by uniform sampling from the meta-BO benchmark environments.

| | Rand. Branin | Camelb. Sin-Noise | Rand. Hartmann6 | GLMNET | RPart | XGBoost |
|---|---|---|---|---|---|---|
| FPACOH-MAP | $\mathbf{0.095 \pm 0.006}$ | $\mathbf{0.046 \pm 0.002}$ | $0.049 \pm 0.003$ | $0.124 \pm 0.010$ | $\mathbf{0.125 \pm 0.006}$ | $\mathbf{0.077 \pm 0.001}$ |
| PACOH-MAP | $0.105 \pm 0.009$ | $0.054 \pm 0.005$ | $0.085 \pm 0.003$ | $0.175 \pm 0.004$ | $0.151 \pm 0.006$ | $0.084 \pm 0.003$ |
| ABLR | $0.180 \pm 0.004$ | $0.049 \pm 0.002$ | $\mathbf{0.044 \pm 0.005}$ | $0.220 \pm 0.005$ | $0.158 \pm 0.010$ | $0.097 \pm 0.004$ |
| NP | $0.146 \pm 0.006$ | $0.053 \pm 0.010$ | $0.063 \pm 0.009$ | $0.202 \pm 0.009$ | $0.176 \pm 0.013$ | $0.197 \pm 0.038$ |
| Learned GP | $0.112 \pm 0.000$ | $0.069 \pm 0.001$ | $0.062 \pm 0.000$ | $0.125 \pm 0.000$ | $0.137 \pm 0.001$ | $0.107 \pm 0.001$ |
| Vanilla GP | $0.123 \pm 0.000$ | $0.085 \pm 0.000$ | $0.089 \pm 0.003$ | $\mathbf{0.123 \pm 0.000}$ | $0.150 \pm 0.000$ | $0.182 \pm 0.000$ |

Table 5: Calibration error of various meta-learned models as well as a Vanilla GP on meta-test tasks generated by uniform sampling from the meta-BO benchmark environments.

experimental findings in Section 5.1 and the results of the BO benchmark studies where F-PACOH performs significantly better than the baselines.

### B.5.2 Offline Meta-Learning

In addition to the simple regret results (see Figure 3), we also provide plots of the inference regret in Figure 5. Note that, since random search does not maintain a machine learning model of the target function, the concept of inference regret does not apply to it. Thus it is not included here.

Overall, the inference regret results in Figure 5 show the same patterns as the simple regret results - F-PACOH significantly outperforms the other methods across all the environments.

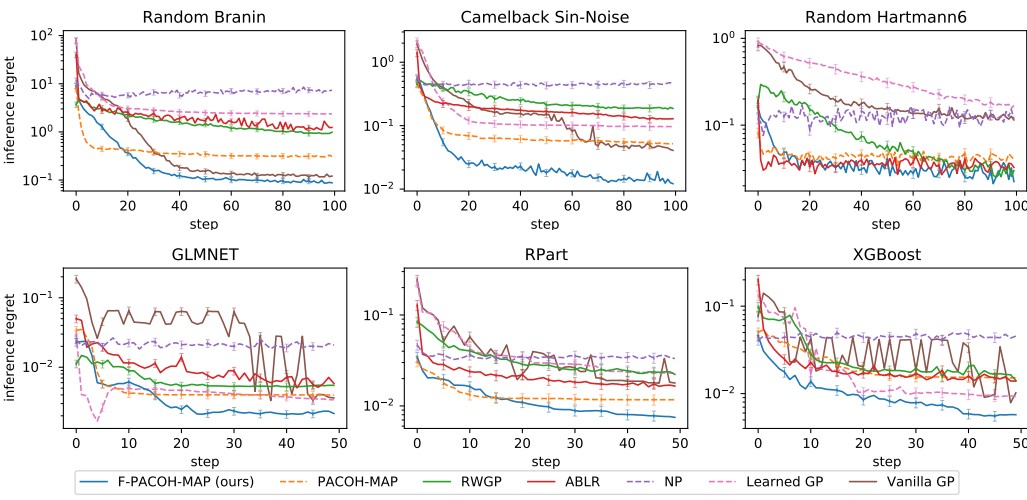

Figure 5: Performance of BO with meta-learned models on simulated function environments (top) and hyper-parameter tuning (bottom). Reported is the inference regret in log-scale, averaged over seeds and function samples, alongside 95% confidence intervals. Consistent with the simple regret, displayed Figure 3, FPACOH significantly outperforms the other methods across all environments.

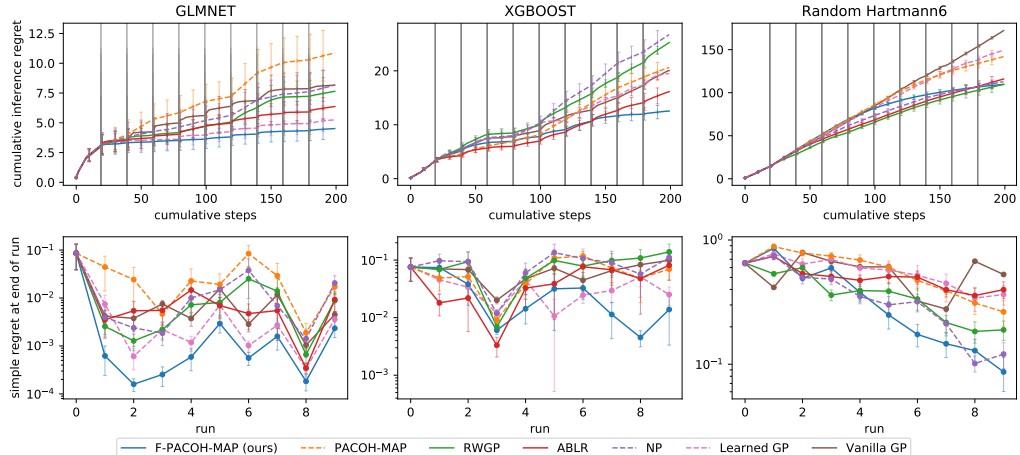

Figure 6: Lifelong BO performance on two simulated function environments and one hyper-parameter tuning benchmark (RPart). The reported results are averages over seeds and random sequences of tasks alongside 95% confidence intervals. While other meta-learners struggle to achieve positive transfer, F-PACOH is able to significantly improve the BO perfomance as it gathers more experience.

### B.5.3 Lifelong Bayesian Optimization

In addition to Figure 4 which displays the the results of our lifelong BO study for three environments, we provide analogous plots for the remaining three environments GLMNET, XGBOOST and Random Hartmann6 in Figure 6. Similar to the previous results, at any time throughout the course of the lifelong BO episode performs among the best and, unlike other methods, keeps improving across the later BO runs.