# OpenReview forum: "Meta-Learning Reliable Priors in the Function Space"
_NeurIPS.cc/2021/Conference — NeurIPS 2021 Poster_

### Official Review · Reviewer_N7j1 · 2021-07-09

**Rating:** 4
**Confidence:** 3

**Summary:**

This paper observes that meta-learning algorithms typically underestimate epistemic uncertainty because of a failure of regularization to respect the structure of meta-learning.
They propose a meta-regularization step that builds on functional KL-divergence to a GP functional prior.
Their experiments in sequential BO suggest that this functional prior outperforms alternatives.

**Limitations And Societal Impact:**

I think the paper would have benefited from more discussion of the limitations of the functional KL estimation.

**Main Review:**

Thank you for this paper, which attacks an important problem for meta-learning: getting good uncertainty estimates that incorporate the 'meta-knowledge' as well as the task-knowledge in a principled way.
The general approach you take makes sense and feels like a promising strategy for dealing with this problem.

Nevertheless, I am not currently recommending acceptance.
I felt that the theoretical contribution is comparatively modest, as the paper mostly combines two previously established tools (adding the functional KL on top of a meta-learning formulation).
Given that, the paper needed to make some real progress on empirical demonstrations showing the usefulness of the approach.
Here, I would have preferred to either see good results on higher-dimensional spaces or some discussion/investigation of failures in high-dimensional spaces.
I would change my mind either if you clarified that I am undervaluing your theoretical contribution, or that the experimental support you offer has broader implications than I currently understand.

# Major comments

One major difficulty with Sun et al.'s approach is the estimation of this supremum over measurement sets.
The approach you take, similar to theirs, is to estimate the expectation over measurement sets (rather than the supremum over a sample).
To the best of my understanding, this might improve stability, but doesn't really match the derivations.
And the estimation of this term is quite difficult in more challenging data-sets.
But if I've misunderstood this, please correct me.

More generally, I worry that the approach of uniformly sampling data to fill in the off-distribution prior will only work in low-dimensional datasets.
This is not something that your experiments are in a position to evaluate.
In order to feel confident that this work is of relevance to the broader NeurIPS community, it would be good to examine settings with more than dozen or so dims (or to explicitly state the limitation if it is not possible).
(My current understanding is that the highest-dim experiment is xgboost with 10 dims, is that right?)
Here, it would be very valuable to push the limits of your approach into domains where it might be harder to get a functional approach to work.

## Originality
I have not seen the use of functional hyper-priors on this formulation of meta-learning before.
However if I have a slight reservation, it is that this paper represents a somewhat limited application to Sun et al.'s Functional VI to Rothfuss et al.'s PACOH.
It would be helpful to build further on these ideas, perhaps by analysing the strengths and limitations of the approach in different settings.

## Quality and Clarity
I felt that the writing could have been much more concise and that this would have improved the presentation of ideas.

I am slightly confused by how uncorrelated the ranking of the non-F-PACOH methods in Figure 3 is.
I think this either means that all the other methods are very sensitive to the dataset in their performance or that they might not be fully tuned to the task?

## Significance
This work could be significant in offering new strategies for dealing with uncertainty in meta-learning.
I suspect that the work will have much greater impact if it is able to apply to a broader range of data types.

# Minor remarks
Abstract could probably be cut significantly without losing meaning. E.g., the first sentence is probably not necessary and second could be "Meta-learning can improve accuracy when data are scarce, but existing methods have unreliable uncertainty which is often overconfident."
More generally, it felt like you could cut a lot of text in a way that would clarify the arguments and contributions.

Line 174 you are missing the KL divergence symbol inside your expectation.

**Time Spent Reviewing:**

2

---

> ### Author Response · Authors · 2021-08-10
> **Authors' response**
>
> We thank you for your assessing our work and suggesting improvements. In the following, we answer your questions and concerns in more detail.
>
> > Here, I would have preferred to either see good results on higher-dimensional spaces or some discussion/investigation of failures in high-dimensional spaces. I would change my mind either if you clarified that I am undervaluing your theoretical contribution, or that the experimental support you offer has broader implications than I currently understand.
>
> Your concerns about high-dimensional domains are absolutely justified. Indeed, we do not expect the proposed approximations of the KL-divergence between stochastic processes to work well for high-dimensional data (d > 50) such as images. In addition, it is unclear what a good hyper-prior for high-dimensional data would look like. For instance, a Gaussian Process with SE kernel would not convey useful inductive bias on the meta-level for data such as images or natural language.
>
> We agree that these limitations and considerations have not been properly discussed in the submitted version of the paper. For this reason, we have added a paragraph to Section 4 of the updated paper where we discuss the method’s limitation to lower-dimensional domains.
>
> At the same time we want to emphasize that our proposed method was devised with particular focus on meta-learning reliable priors for interactive machine learning, e.g. Bayesian optimization (BO). In this context, data is very scarce and the domain of feasible problems rarely exceeds 10-20 dimensions. Our empirical evaluations feature realistic BO experiments with domains of up to 10 dimensions (see Table 1 in the Appendix) which is already quite high-dimensional for a BO problem.
>
> Compared to other popular meta-learners, F-PACOH is very sample efficient. It’s key advantage over popular methods such as Neural Processes (NPs) and MAML is that it is still able to produce reliable results, even when there are only a handful of meta-training tasks with few data points available. To be specific, in the results of the BO experiments in Figure 3 and the supervised learning experiments in Appendix B.5.1, F-PACOH always outperforms NPs by a substantial margin. Most importantly, as we demonstrate in the life-long BO experiments, F-PACOH is able to succeed in highly challenging scenarios such as meta-training with one single task and non-i.i.d. data points, where other popular meta-learners such as NPs fail.
>
> Thus, F-PACOH is highly relevant for application domains such as robotics, biology and AutoML, where data and tasks are scarce and costly to obtain. With the aim to demonstrate the practical relevance of our approach, we put in a lot of effort to provide the extensive AutoML experiments in the paper which showcase how F-PACOH can be used for tuning machine learning models more efficiently and potentially save thousands of hours of compute. Moreover, we are currently collaborating with an industry partner, using F-PACOH + BO for tuning the controllers of high-precision robots more efficiently.
>
> In summary, our work closes a gap in the landscape of existing meta-learning approaches by providing an algorithm that is able to provide reliable uncertainty estimates in scenarios with only a few tasks and non-i.i.d. data. By its very nature, F-PACOH is limited to lower-dimensional data domains. Nonetheless, there are many challenging real-world problems that are not high-dimensional and require dealing with small amounts of non-i.i.d meta-training data. These are the problems F-PACOH is meant to and, as we show, also successfully able to solve. Thus, we consider our proposed method both relevant for the machine learning community as well as for practitioners in the industry and sciences.
> In light of this, we kindly ask you to reconsider your overall rating of the paper.
>
> >  estimat[ing] the expectation over measurement sets (rather than the supremum over a sample) [...] might improve stability, but doesn't really match the derivations. And the estimation of this term is quite difficult in more challenging data-sets
>
> Indeed, your assessment is pretty accurate. Directly estimating the actual KL-divergence between stochastic processes (i.e. the sup over finite measurement sets) is a very hard and unsolved problem in research. One possibility to relax this hard problem is to constrain the cardinality of the measurement set to a fixed size of $m$ points, i.e. $\sup_{\mathbf{X} \in \mathcal{X}^m} KL[q(\mathbf{h}^\mathbf{X}) || \rho( \mathbf{h}^\mathbf{X})]$. When comparing this to the expectation over measurement sets $\mathbb{E}_{\mathbf{X}_i}  \left[KL[ q(\mathbf{h}^{\mathbf{X}_i})||\rho(\mathbf{h}^{\mathbf{X}_i}) ]\right]$, if the sampling distribution for the measurement sets $\mathcal{X}_i^M$ of size $m$ has full support over the domain, the prior that minimizes the F-PACOH-MAP objective with expectation over measurement sets is the same as the prior that minimizes the objective with the sup over measurement sets of fixed size $m$.
> This is an adaptation of Corollary 3 in Sun et al. (2019). While, in theory, both relaxations of the functional KL lead to the same minimizer, optimizing the expectation over measurement sets is much easier, more stable and empirically works better.
>
> Does this help to clarify the connection between the supremum and the expectation over measurement sets?
> At some point we had a discussion thereof in the paper but removed it due to space constraints. What do you suggest? Should we put the discussion back into the updated version of the paper, now that we have one page more space?
>
> > I am slightly confused by how uncorrelated the ranking of the non-F-PACOH methods in Figure 3 is. I think this either means that all the other methods are very sensitive to the dataset in their performance or that they might not be fully tuned to the task?
>
> As we describe in Appendix B.4., we have tuned the hyper-parameters of each method / baseline on each of the benchmarks individually with random search. The hyper-parameter search ranges and all other details are specified in the code which we link to in Appendix B.3. The selected hyper-parameters are all reported in our [experiment repository](https://www.dropbox.com/sh/n2thesjq87sh66j/AACg-HKMl1NhQpaMOHvvUEOfa?dl=0) (also provided in Appendix B4). Given that all the methods have been thoroughly tuned to the BO environments, the best explanation is that some of the baseline methods are sensitive to the environment. This is not surprising, given the fact that most of the baselines do not properly regularize on the meta-level and thus are very prone to meta-overfitting.
>
> > Abstract could probably be cut significantly without losing meaning.
>
> Thanks for the detailed suggestions for making the abstract more concise. We have significantly cut the length of the abstract and in general.
>
> > I felt that the writing could have been much more concise and that this would have improved the presentation of ideas.
>
> We went again through the paper and tried to make the arguments more concise, freeing up space which we used to incorporate a better discussion of the limitations, e.g. to high-dimensional data.

---

> > ### Comment · Reviewer_N7j1 · 2021-09-01
> > **Thank you for your response**
> >
> > Thanks for your response. Unfortunately, I don't believe this changes my evaluation of the relative novelty of the method, which is a relatively straightforward extension of two related methods. This doesn't change the fact that your paper is well-executed.
> >
> > I understand that you are motivated primarily by BO, and I do think this is an important problem. There is no difficulty at all in a paper focusing primarily on small spaces. Rather, my concern is that the Functional KL paper already establishes the effectiveness of their method in low-dim settings, and this paper mostly applies that technique in a meta-learning context. I do think there is a lot to like about this paper, but that it falls below the threshold for NeurIPS of significance and novelty. Other reviewers could reasonably disagree with me, and they may well do so!

---

> ### Author Response · Authors · 2021-08-24
> **Did we address all your concerns and questions?**
>
> Dear Reviewer,
>
> Since the discussion period is coming to an end soon, we want to check back if we were able address all your concerns and questions with our detailed response below. Do you still have some questions or concerns about the paper? In your review you state that you are open to change your mind. Considering that we have improved the paper based on your feedback and clarified the contribution / importance of the proposed method, are you increasing your rating of the paper?
>
> Thanks a lot!

---

### Official Review · Reviewer_iGTP · 2021-07-09

**Rating:** 6
**Confidence:** 3

**Summary:**

This work introdcues a new way to regularize priors in meta-learning through the use of stochastic processes as hyper-priors. The authors rely on sampling-based measurements to estiamte the KL divergence between the meta-learned prior and a stochastic process hyper-prior.

Through experiments on toy datasets, the algorithm, F-PACOH, demonstrates principled uncertainty estimates. It provides higher levels of uncertainty far from training data, which is in contrast to previous methods for imposing meta-learning priors and hyper-priors. Finally, the authors apply their method to a Bayesian Optimization meta-learning problem and demonstrate decreased regret compared to other methods benchmarked.

**Limitations And Societal Impact:**

The authors didn't spend much time on the limitations of their work, beyond addressing the cubic runtime of their methods. Some things that may or may not be limitations that could be addressed:
* Are you always able to get good sampling estimates for the KL-divergence and its gradient?
* Can this method scale in parallel at all? Can it handle large batches of evaluations?

Additionally, societal impacts weren't addressed. Could you talk about how BO can develop novel therapeutics through protein optimization, for instance?

**Main Review:**

In my opinion, this paper is a strong combination of several existing ideas in meta-learning and stochastic processes. The application of KL divergence sampling-based estimates from Sun et al. in conjunction with the PACOH framework demonstrated principled uncertainty estimates and strong performance on Bayesian optimization tasks. The combination of these ideas yielded a method that has strong empirical justifications and theoretical motivations.

Throughout, the authors were very clear in their presentation. One thing that may improve clarity is an appendix section (didn't have time to check those, so sorry if this already exists) on stochastic processes. I consulted Sun et al. for a brief review and that was helpful, but making the paper more self contained for someone coming from an adjacent area could be useful. Another point to that effect is that the authors keep referring to "regularization in the function space" and my brain keeps asking "the function space of <what>?" That could just be how one refers to function spaces but a few examples of writing "the function space of the prior" would have defintely helped me get the concept quicker.

One error (I believe) is on Line 174. The authors missed the "KL" in the expression $$\mathbb{E}_{X_i}KL[q(h^{X_i}||\rho (h^{X_i}))]$$

In terms of experiments, I would be interested in more demonstrations that F-PACOH generates principled uncertainty estimates. The 1-D toy datasets did show that property but more empirical validation would be welcome. I'd be particularly interested in uncertainty estimates on out-of-distribution data that has undergone dataset shift (i.e. Ovadia et al.).

Ultimately, my impression is that this paper represents a solid advance in (hyper-)priors for metalearning and combines several ideas into a useful new method. Given it's strong performance on the BO tasks, I could see other researchers building on these ideas.

Update after responses:
I think that a major limitation of the work is its restriction to lower dimensional settings. This will fundamentally limit its applications and is a larger issue than I originally thought. I still think this is a good paper but am slightly lowering my score because of this.

**Time Spent Reviewing:**

3

---

> ### Author Response · Authors · 2021-08-10
> **Authors' response**
>
> Thank you for your encouraging feedback and your useful suggestions for improving the paper. In the following, we address your concerns and questions one by one:
>
> > One thing that may improve clarity is an appendix section [...] on stochastic processes.
>
> Following your suggestion, we have added a small section at the beginning of the Appendix on stochastic processes for readers that are unfamiliar with the topic.
>
> > I would be interested in more demonstrations that F-PACOH generates principled uncertainty estimates. The 1-D toy datasets did show that property but more empirical validation would be welcome.
>
> In the paper, we actually directly assess the uncertainty estimates on all of the 6 BO environments which are introduced in Section 5.2. In particular, Table 5 in the appendix reports the calibration error (see Eq. 26 for definition) for each of the methods and BO environments. In terms of the calibrations of its uncertainty estimates, F-PACOH significantly outperforms the other methods in the majority of the environments. We acknowledge that Table 5 is never properly mentioned in the main part of the paper. For this reason, we have added two more sentences to Section 5.1., referring to and discussing the results in Table 5.
>
> > I'd be particularly interested in uncertainty estimates on out-of-distribution data that has undergone dataset shift (i.e. Ovadia et al.)
>
> The data which is used for computing the calibration plots and calibration errors is collected with GP-UCB. Thus, the data is non-i.i.d. and, depending on the task, the BO algorithm evaluates certain regions of the domain more densely than others, while, over time, increasingly concentrating around the maximum. Thus, a certain amount of distributional shift is already part of our evaluation procedure.
>
> > Some things that may or may not be limitations that could be addressed: Are you always able to get good sampling estimates for the KL-divergence and its gradient?
>
> The quality of sampling estimates of the KL-divergence degrades with the dimensionality of the data domain $\mathcal{X}$. Thus, we do not expect our method to work on high-dimensional data such as images. We agree that this limitation has not been properly discussed in the submitted version of the paper. For this reason, we have added a paragraph to Section 4 of the updated paper where we discuss the method’s limitation to lower-dimensional domains.
>
> > Additionally, societal impacts weren't addressed. Could you talk about how BO can develop novel therapeutics through protein optimization, for instance?
>
> Thank you for pointing out potential applications of our method. We have added a small broader impact section after the conclusion which discusses such applications and potential societal impact.

---

> > ### Comment · Reviewer_iGTP · 2021-08-30
> > **re:**
> >
> > Hi authors,
> >
> > Thank you for responding to my questions -- I appreciate the work you put in! I'm going to update my review based on your responses.

---

### Official Review · Reviewer_RiXp · 2021-07-16

**Rating:** 6
**Confidence:** 4

**Summary:**

This paper discusses a meta-learning method with priors defined on function spaces. Building on PACOH, a meta-learning framework with PAC-Bayesian bounds, the paper proposes F-PACOH where the hyper-priors and priors defining meta-learning models in function spaces. In original PACOH, the hyper-priors are set to be parametric distributions (usually Gaussians), and the authors argue that such parametric priors cannot effectively capture the functional structure of the tasks that we ultimately want to enforce for meta-learners. On the other hand, functional priors directly regularize functions of interest to be close to prior knowledge, e.g., standard Gaussian processes, so result in more reliable uncertainty estimates.

The inference is done by optimizing the PAC-Bayesian bound, and the caveat here is the computation of KL-divergence between functional hyper-priors and hyper-posteriors. This can be approximated by the technique introduced in Sun et al., 2021, where a set of measurements points are sampled and the KL-divergence is approximated with the evaluations on those measurement points.

The paper mainly focuses on the application of the proposed method to Bayesian optimization. On several benchmarks, the proposed F-PACOH is demonstrated to quickly meta-learn and improves on similar BO tasks.

**Limitations And Societal Impact:**

The authors discussed both limitation and potential negative societal impact of the proposed method.

**Main Review:**

Overall, the paper is clearly written and easy to follow. The motivation is well set, and the idea to use functional priors is a reasonable solution to the problem. The BO experiments, especially the sequential BO, are interesting. I'm no expert in BO literature, so my assessment may be wrong, but at least for me, these experiments seem novel.

I cannot say the proposed method is novel enough so that the paper should be accepted solely based on its novelty; the paper is definitely well-executed, but the core contribution is rather an incremental combination of well-established frameworks; PACOH and functional variational Bayesian neural nets (FBNNs). Roughly speaking, F-PACOH is an FBNN version of PACOH which uses the same inference techniques as the original FBNNs. The application to meta-learning is new, but it does not incur additional technical difficulty.

Although the framework is generic so virtually one can apply any functional hyper-priors and priors, the paper only presents the case when both of them and priors are set to be Gaussian processes (GPs). GP is easy to deal with due to its closed-form marginals. Have you actually tested with other types of priors or hyper-priors, for instance, implicit processes? In such a case, as noted in the paper, the gradient of KL-divergence should be approximated via SSGE or sliced score matching. Personally, I find the gradient estimates with such techniques unreliable, so wonder if the proposed method works well with them.

The measurement points should be sampled for the KL-divergence approximation. This might be trivial for 1D real line for instance, but what if the target domain is high-dimensional space such as images or sequences? Would the uniform sample method still work well? I guess this would be important because the proposed meta-learning will be more useful for such a high-dimensional setting for which the performance of a vanilla GP based BO is expected to degrade.

In the experiments, a straightforward baseline I can think of is, during meta-training gather all the training task points  $D_{1:T}$,
construct a posterior GP conditioned on $D_{1:T}$, and use it as a hyper-prior for subsequent tasks. I guess this can be a strong baseline despite having to store all the training task data $D_{1:T}$.

It would be good to directly assess the uncertainty estimates for BO tasks as well (with calibration error plots for instance).

How sample efficient the proposed method is? For instance, how are the performances of the algorithms w.r.t. different numbers of training tasks? As far as I know, NP requires quite many tasks to be reliably trained, even for simple GP regression tasks. Judging from the simulation results where F-PACOH is trained with only 10 tasks and still is able to reliably adapt to new tasks, I think F-PACOH to be much more sample-efficient than NP.


**Time Spent Reviewing:**

6 hours

---

> ### Author Response · Authors · 2021-08-10
> **Authors' response**
>
> Thank you for taking the time to engage with the paper and provide a thorough review. We are happy to hear that you enjoyed reading our paper and acknowledge the novelty and contribution of BO experiments.  In the following, we are addressing your questions and concerns one by one.
>
> > Have you actually tested with other types of priors or hyper-priors [...]? In such a case [...] the gradient of KL-divergence should be approximated via SSGE or sliced score matching. Personally, I find the gradient estimates with such techniques unreliable, so wonder if the proposed method works well with them.
>
> In the meantime, we have run experiments where we use Monte Carlo estimation of the KL divergence and SSGE to estimate the KL divergence. If we change the hyper-prior it is hard to assess to what extent the observed performance difference is due to the score approximation as opposed to different hyper-prior. For this reason, we have used the same GP hyper-prior as in the other experiments, but do not give access to its finite marginals. This way all the performance difference stems from the MC and SSGE approximations.
>
> Empirically, the SSGE version of F-PACOH only performs slightly worse than the one which uses the closed form KL divergence and still yields significantly lower regrets than the other baselines. In the following, we give specific examples of the simple regret at step 50 in the offline meta-training setting (Section 5.3.) which can be compared to Figure 3 in the paper: GLMNET: $0.000705$, RPart: $008166$, XGBOOST: $0.004541$.
> In particular, we have used a measurement set of 20 points and 500 samples from $\rho(\mathbf{h}^{\mathbf{X}})$ to estimate the hyper-prior score with SSGE. With these settings, the SSGE version of F-PACOH runs 2-3 times slower than the one with the closed-form KL and the same measurement set size.
>
> Overall, we shared your concern that the score estimates would be unreliable and still don’t think that the individual gradient estimates are good. However, we conjecture that, since a new score estimate is formed in each iteration it can be thought of as similar to stochastic gradient descent, where the randomness of the gradient estimates average out over many iterations, leading to good results nonetheless.
>
> Would you like us to add a small subsection with these results to the main body or the appendix of the paper?
>
> > The measurement points should be sampled for the KL-divergence approximation. This might be trivial for 1D real line for instance, but what if the target domain is high-dimensional space such as images or sequences? Would the uniform sample method still work well?
>
> Your concerns about high-dimensional domains are absolutely justified. Indeed, we do not expect the proposed approximations of the KL-divergence between stochastic processes to work well for high-dimensional data such as images. Our proposed method was devised with particular focus on meta-learning reliable priors for BO. In this context, data is very scarce and the dimensionality of feasible problems rarely exceeds 20.
>
> Unlike in the supervised learning setting, the previously collected data is non-i.i.d. and the BO algorithm actively chooses the evaluation points $x \in \mathcal{X}$ as opposed to having a fixed data distribution $p(x)$. Since we do not know in advance which points in the domain will be evaluated by the BO algorithm, we want the prior to be regularized by the hyper-prior uniformly throughout the entire domain to get reliable uncertainty estimates also beyond the support of the meta-training data. However, in a supervised learning setting it may be better to form a conservative estimate of the data distribution $p(x)$ and use this for sampling the measurement sets.
>
> We agree that these limitations and considerations have not been properly discussed in the submitted version of the paper. For this reason, we have added a paragraph to the updated paper where we discuss method’s limitation to lower-dimensional domains and alternatives to uniform sampling.
>
> > It would be good to directly assess the uncertainty estimates for BO tasks as well (with calibration error plots for instance)
>
> We actually do assess directly the uncertainty estimates for the BO tasks. In particular, Table 5 in the appendix reports the calibration error (see Eq. 26 for definition) for each of the methods and BO environments. In terms of the calibrations of its uncertainty estimates, F-PACOH significantly outperforms the other methods in the majority of the environments. We acknowledge that Table 5 is never properly mentioned in the main part of the paper. For this reason, we have added two more sentences to Section 5.1., referring to and discussing the results in Table 5.
>
> > How sample efficient the proposed method is? For instance, how are the performances of the algorithms w.r.t. different numbers of training tasks? As far as I know, NP requires quite many tasks to be reliably trained, even for simple GP regression tasks. Judging from the simulation results where F-PACOH is trained with only 10 tasks and still is able to reliably adapt to new tasks, I think F-PACOH to be much more sample-efficient than NP.
>
> Compared to other popular meta-learners, F-PACOH is very sample efficient. It’s key advantage over methods such as Neural Processes (NPs) and Bayesian MAML (BMAML) is that it is still able to produce reliable results, even when there are only a handful of meta-training tasks with a few data points available. This makes it so relevant for interactive machine learning problems where we start with very little data and actively gather more and more data to improve the performance (e.g. active learning or BO).
>
> To be specific, in our experiments, we used only 10 - 30 tasks and 10 - 100 data points per task for meta-training (see Table 1 in the appendix).  As we can see in the results of the BO experiments in Figure 3, as well as the supervised learning experiments in Appendix B.5.1, F-PACOH always outperforms NPs by a substantial margin. Thus, F-PACOH is highly relevant for application domains such as robotics and biology where data and tasks are scarce and costly to obtain. On the other hand, in settings where there are high-dimensional data and thousands of tasks for meta-training, F-PACOH may no longer perform competitively and other methods such as NPs may be the better choice.

---

> > ### Comment · Reviewer_RiXp · 2021-08-25
> > **thanks**
> >
> > I appreciate the author's response which resolved most of my concerns. Still, I agree with the other reviewers' thoughts on the novelty, so I keep my score unchanged.

---

> ### Author Response · Authors · 2021-08-24
> **Did we address all your concerns and questions?**
>
> Dear Reviewer,
>
> Since the discussion period is coming to an end soon, we want to check back if we were able address all your concerns and questions. Do you still have some questions or concerns about the paper which are unaddressed?
>
> In response to your feedback, we have
> * added a discussion of the method's limitation to lower-dimenional domains,
> * included experiment results were the hyper-prior's score is estimated via SSGE,
> * better point out in Section 5.1 that a an calibration assessment for all of the environments and methods is already included in the appendix.
>
> Given these improvements and our clarifications to your questions / concerns in our rebuttal response below, do you consider increasing your score?
>
> Thanks a lot!

---

### Official Review · Reviewer_VyPu · 2021-08-01

**Rating:** 7
**Confidence:** 3

**Summary:**

This paper considers improving uncertainty estimations for meta-learning. Previous work [1] has done this via using a hyper-prior over the prior parameters for meta-regularization. This hyper-prior, however, took the form of a simple Gaussian distribution over the prior parameters, which imposes smoothness regularization on the prior but may not be ideal for enforcing uncertainty in regions where no meta-training data is available. To remedy this, the authors propose a hyper-prior in the function space in the form of a stochastic process. They then show how the meta-learning algorithm can be defined so that it uses stochastic processes for the prior, hyper-prior and hyper-posterior. The experiments are conducted on a Bayesian optimization setup, where meta-training involves using a sequences of related BO problems corresponding to n target functions and evaluation involves using the meta-learned model to perform BO on a new target function. For example, one such setting is hyper-parameter optimization for machine learning models, where the training task functions correspond to training and testing the machine learning model on different datasets. Across different benchmarks in this setting, they show their proposed algorithm performs better than [1].

[1] Rothfuss et al. PACOH: Bayes-optimal meta-learning with PAC-guarantees. ICML 2021.

**Ethical Concerns:**

No ethical concerns.

**Limitations And Societal Impact:**

Not explicitly discussed but should not be different than existing work in this area of meta-learning.

**Main Review:**

Originality: the paper is proposing a modification to [1] to improve its behavior and performance; however, I believe the proposed change is interesting and does seem to lead to better performance.

Quality: the paper seems technically sound. There was strong justification provided for the proposed method and the experimental results seem to show that it does provide the type of benefit in uncertainty estimation that was discussed in the motivation for the work.

Clarity: the paper is very well-written. Explains the proposed method and its justifications well and the experiments are thoroughly described.

Significance: I think Bayesian optimization is an interesting application of meta-learning and the experiments & benchmarks considered for meta-learning for Bayesian optimization in this work could be interesting to other researchers and help build more momentum in this area.

[1] Rothfuss et al. PACOH: Bayes-optimal meta-learning with PAC-guarantees. ICML 2021.

**Time Spent Reviewing:**

4

---

> ### Author Response · Authors · 2021-08-10
> **Authors' response**
>
> Thank you for your positive assessment. We are happy to hear that you consider the proposed method relevant, sound and of interest to other researchers. We will be happy to clarify further questions in case they arise during the discussion period.

---

### Author Response · Authors · 2021-08-10
**General response to all reviewers**

We would like to thank all of the reviewers for their careful reading, assessment, and feedback. We are pleased that the majority of the reviewers finds the paper to be well-/clearly written [VyPu, RiXp, iGTP] and that the proposed method “has strong empirical justifications and theoretical motivations” [iGTP]. In particular, we are excited to hear that most of the reviewers recognize the novelty and contribution of our extensive meta-learning for BO experiments [VyPu, RiXp, iGTP] and consider them “interesting to other researchers [as they could] help [to] build more momentum in this area” [VyPu].

We agree with the reviewers’ concern about high-dimensional data. For this reason, we have added a discussion of the method’s limitations to lower-dimensional data domains to the updated version of the paper [RiXp, N7j1].

In light of this, we want to emphasize that our paper aims to close a gap in the landscape of existing meta-learning approaches by providing an algorithm that is able to provide reliable uncertainty estimates in scenarios with only a handful of tasks and small amounts of non-i.i.d. data. As we try to showcase with our AutoML experiments, we believe that there are many relevant real world problems that are not high-dimensional and can benefit from the proposed F-PACOH approach - especially in combination with interactive systems such as Bayesian optimization, active learning and reinforcement learning.

---

### Decision · Program_Chairs · 2021-09-27

**Decision:**

Accept (Poster)

**Comment:**

This paper uses PAC-Bayes meta-learning to do meta-learning using the functional KL.  They demonstrate this in a series of experiments where they meta-learn hyperpriors of Gaussian processes for the purpose of doing Bayesian optimization among related tasks.

The reviewers thought the paper was well written, technically strong and interesting.  The review scores were 6, 7, 4, 6.  The main concern here seems to be novelty.  The reviewer arguing for reject expressed that they thought it was a straightforward extension of functional KL and meta-learning ("The combination of functional-KL and meta-learning problem seem fairly marginal.”).

One major criticism shared by all reviewers seems to be the scale of the experiments, in that they’re limited to low-dimensional problems due to limitations of the function space view.  One reviewer lowered their score from a 7 to a 6 as a result of this during the discussion period. It seems that the authors are aware of this concern and have promised to address this in their discussion.  From the authors: “We agree with the reviewers’ concern about high-dimensional data. For this reason, we have added a discussion of the method’s limitations to lower-dimensional data domains to the updated version of the paper.”

The majority vote among the reviewers was to accept the paper, citing that the paper is well written, technically well-executed and empirically strong (although limited to low-D problems). The application to meta-Bayesian optimization seems well motivated and practically useful.  Thus the recommendation is to accept the paper as a poster.  Hopefully the reviewers feedback can be incorporated into the camera ready version (including discussion of low-D limitations) to make the paper stronger.